# EFFICIENT AND SCALABLE REINFORCEMENT LEARNING VIA HYPERMODEL

## ABSTRACT

Data-efficient reinforcement learning(RL) requires deep exploration. Thompson sampling is a principled method for deep exploration in reinforcement learning. However, Thompson sampling need to track the degree of uncertainty by maintaining the posterior distribution of models, which is computationally feasible only in simple environments with restrictive assumptions. A key problem in modern RL is how to develop data and computation efficient algorithm that is scalable to large-scale complex environments. We develop a principled framework, called HyperFQI, to tackle both the computation and data efficiency issues. HyperFQI can be regarded as approximate Thompson sampling for reinforcement learning based on hypermodel. Hypermodel in this context serves as the role for uncertainty estimation of action-value function. HyperFQI demonstrates its ability for efficient and scalable deep exploration in DeepSea benchmark with large state space. HyperFQI also achieves super-human performance in Atari benchmark with 2M interactions with low computation costs. We also give a rigorous performance analysis for the proposed method, justifying its computation and data efficiency. To the best of knowledge, this is the first principled RL algorithm that is provably efficient and also practically scalable to complex environments such as Arcade learning environment that requires deep networks for pixel-based control.

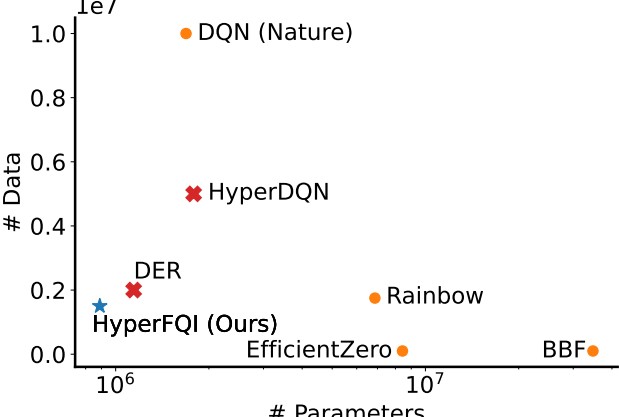

Figure 1: This figure investigates the relationship between the required training data and the model parameters for achieving human-level performance using various algorithms. The evaluation is conducted using IQM Agarwal et al. (2021) on 26 Atari games. The $x$-axis represents the amount of training data required to achieve human-level performance, measured in 1.0 IQM. The $\times$ indicates that the algorithm fails to reach human-level performance with the corresponding amount of training data. The $\star$ denotes that our proposed algorithm, HyperFQI, achieves human-level performance with minimal parameters and relatively little training data.

## 1 INTRODUCTION

In reinforcement learning (RL), intelligent exploration relies on decisions that are driven not only by expectations but also by epistemic uncertainty (Osband et al., 2019b). Actions are taken to resolve

epistemic uncertainty not only based on immediate consequences but also on what will be observed over subsequent time periods, a concept known as deep exploration (Osband et al., 2019b). One popular exploration scheme in RL is Thompson Sampling (TS), which makes decisions based on a posterior distribution over models (Thompson, 1933; Russo et al., 2018). A basic form of TS involves sampling a model from the posterior and selecting an action that optimizes the sampled model.

However, generating exact posterior samples is computationally tractable only for simple environments, such as tabular MDPs with Dirichlet priors over transition probability vectors (Osband et al., 2013). For complex domains, approximations are necessary (Russo et al., 2018). In order to address this need, Osband et al. (2019b) developed randomized least-squares value iteration (RLSVI). RLSVI aims to approximate sampling from the posterior over the optimal value function without explicitly representing the distribution. The algorithm achieves this by randomly perturbing a prior and an accumulated dataset and fitting a point estimate of the value function to this perturbed prior and data. The induced randomness from these perturbations leads to deep exploration, improving data efficiency (Osband et al., 2019b).

While RLSVI avoids the explicit maintenance of a posterior distribution, it still requires computationally intensive operations to generate a new point estimate for each episode. These computations do not leverage previously computed point estimates and therefore cannot be incrementally updated. Ensemble sampling has been proposed as an alternative approach to approximate RLSVI's performance. It involves maintaining a set of point estimates, with each estimate updated incrementally as data accumulates (Osband et al., 2016; 2019b). Nevertheless, maintaining an ensemble of complex models can be computationally burdensome. Furthermore, to obtain a good approximation of the posterior distribution, the ensemble size needs to grow significantly with the complexity of the distribution (Dwaracherla et al., 2020; Osband et al., 2021; Li et al., 2022a; Qin et al., 2022).

Alternatively, instead of maintaining an ensemble of models, one can learn a hypermodel. A hypermodel can be used to generate approximate posterior samples, as discussed in prior works (Dwaracherla et al., 2020; Li et al., 2022a). This approach shows promise, but it requires a representation that can be more complex than a point estimate of the value function. The computational requirements and the number of parameters needed for this representation, however, lack theoretical understanding.

None of these algorithms have been shown to be computationally efficient, data efficient and scalable at the same time. In particular, RLSVI is data efficient but it is neither computationally efficient nor scalable. Ensemble Osband et al. (2016) or previous hypermodel-related approaches (Li et al., 2022a) are computationally tractable to scale to complex environment with deep networks but they are not data efficient enough and also have no theoretical guarantees. This paper aims to develop a principled RL algorithm that is both computationally and data efficient and also scalable to large-scale complex environments.

## 1.1 CONTRIBUTIONS

We propose a novel algorithm, called HyperFQI, that combines the benefits of RLSVI and hypermodel-based approaches. HyperFQI is based on the fitted Q-iteration (FQI) algorithm (Ernst et al., 2005; Mnih et al., 2015), which is a batch RL algorithm that learns a Q-function by fitting a regression model to a dataset of state-action-reward-state tuples. HyperFQI maintains a hypermodel that can be used to generate approximate posterior samples and carefully designs a way to sample from hypermodel for both training and action selection.

- HyperFQI is the first algorithm in the literature solving DeepSea at a large scale up to $100^2$ states, see details in Section 4.1.
- HyperFQI is also the first one achieving human-level performance in Atari games, when considering both data and computation efficiency, see details in Figure 1 and Section 4.2.
- We provide a rigorous performance analysis for the proposed method in Section 5, justifying its computation and data efficiency. HyperFQI achieves the Bayesian regret bound of order $\tilde{O}(H^2\sqrt{|\mathcal{X}||\mathcal{A}|K})$ for finite horizon MDPs with $H$ horizon, $|\mathcal{X}|$ states, $|\mathcal{A}|$ actions, and $K$ episodes, sharing the same order with famous RLSVL (Osband et al., 2019b) and PSRL (Osband & Van Roy, 2017). The additional computation burden of HyperFQI than

single point estimate is only logarithmic in $|\mathcal{X}|$ and $|\mathcal{A}|$ and episode number $K$, i.e. the additional dimension is $M = \tilde{O}(\log(|\mathcal{X}||\mathcal{A}|K))$. The analysis is enabled by our novel probability tools in Appendix G, which maybe of independent interest.

To the best of knowledge, this is the first principled RL algorithm that is provably efficient and also practically efficiently scalable to complex environments such as DeepSea with large state space and Arcade learning environment. We believe this work serves as a bridge for theory and practice in reinforcement learning.

## 2 PRELIMINARY

### 2.1 REINFORCEMENT LEARNING

We consider the episodic RL setting in which an agents interacts with an unknown environment over a sequence of episodes. We model the environment as a Markov decision problem (MDP) $M = (\mathcal{S}, \mathcal{A}, R, P, s_{\text{terminal}}, \rho)$, where $\mathcal{S}$ is the state space, $\mathcal{A}$ is the action space, $\text{terminal} \in \mathcal{S}$ is the terminal state, and $\rho$ is the initial state distribution. For each episode, the initial state $S_0$ is drawn from the distribution $\rho$. In each time period $t = 1, 2, \ldots$ within an episode, the agent observes a state $S_t \in \mathcal{S}$. If $S_t \neq s_{\text{terminal}}$, the agent selects an action $A_t \in \mathcal{A}$, receives a reward $R_{t+1} \sim R(\cdot \mid S_t, A_t)$, and transitions to a new state $S_{t+1} \sim P(\cdot \mid S_t, A_t)$. An episode terminates once the agent arrives at the terminal state. Let $\tau$ be the termination time of a generic episode, i.e., $S_\tau = s_{\text{terminal}}$. Note that $\tau$ is a stopping time in general. To illustrate, we denote the sequence of observations in episode $k$ by $\mathcal{O}_k = (S_{k,0}, A_{k,0}, R_{k,1}, \ldots, S_{k,\tau_k-1}, A_{k,\tau_k-1}, R_{k,\tau_k})$ where $S_{k,t}, A_{k,t}, R_{k,t+1}$ are the state, action, reward observed at $t$-th time period of the $k$-th episode and $\tau_k$ is the termination time at episode $k$. We denote the history of observations made prior to episode $k$ by $\mathcal{H}_k = (\mathcal{O}_1, \ldots, \mathcal{O}_{k-1})$.

A policy $\pi : \mathcal{S} \to \mathcal{A}$ maps a state $s \in \mathcal{S}$ to an action $a \in \mathcal{A}$. For each MDP $M$ with state space $\mathcal{S}$ and action space $\mathcal{A}$, and each policy $\pi$, we define the associated state-action value function as:

$$Q_M^\pi(s,a) := \mathbb{E}_{M,\pi}\left[\sum_{t=1}^{\tau} R_t \mid S_0 = s, A_0 = a\right]$$

where the subscript $\pi$ next under the expectation is a shorthand for indicating that actions over the whole time periods are selected according to the policy $\pi$. Let $V_M^\pi(s) := Q_M^\pi(s, \pi(s))$. We say a policy $\pi^M$ is optimal for the MDP $M$ if $\pi^M(s) \in \arg\max_\pi V_M^\pi(s)$ for all $s \in \mathcal{S}$. To simplify the exposition, we assume that under any MDP $M$ and any policy $\pi$, the termination time $\tau < \infty$ is finite with probability 1.

The agent is given knowledge about $\mathcal{S}, \mathcal{A}, s_{\text{terminal}}$, and $\rho$, but is uncertain about $R$ and $P$. The unknown MDP $M$, together with reward function $R$ and transition function $P$, are modeled as random variables with a prior belief. The agent's behavior is governed by a RL algorithm $\text{alg}$ which uses the history of observations $\mathcal{H}_k$ to select a policy $\pi_k = \text{alg}(\mathcal{S}, \mathcal{A}, \mathcal{H}_k)$ for the $k$-th episode. The design goal of RL algorithm is to maximize the expected total reward up to episode $K$

$$\mathbb{E}_{[}M, \text{alg}\left[\sum_{k=1}^{K}\sum_{t=1}^{\tau_k} R_{k,t}\right] = \mathbb{E}_{[}M, \text{alg}\left[\sum_{k=1}^{K} V_M^{\pi_k}(s_{k,0})\right]. \tag{1}$$

where the subscript $\text{alg}$ under the expectation indicates that policies are generated through algorithm $\text{alg}$. Note that the expectations in both sides of Equation (1) is over the stochastic transitions and rewards under the MDP $M$, the possible randomization in the learning algorithm $\text{alg}$. The expectation in the LHS of Equation (1) is also over the randomness in the termination time $\tau_k$.

### 2.2 HYPERMODEL

We build RL agents based on the hypermodel framework (Li et al., 2022a; Dwaracherla et al., 2020; Osband et al., 2021). Hypermodel is a function $f$ parameterized with $\theta$, receiving input $x \in \mathbb{R}^d$ and an random index $\xi \in \mathbb{R}^M$ from reference distribution $P_\xi$, making predictions $f_\theta(x, \xi) \in \mathbb{R}$. We aim to capture the uncertainty via the variation over the hypermodel predictions by the random index $\xi$.

Hypermodel parameter $\theta$ is trainable to adjust its uncertainty representation when seeing more data. The reference distribution $P_\xi$ remain fixed throughout the training. For example, linear-Gaussian model is a special case of hypermodel with parameter $\theta = (\boldsymbol{A}, \mu)$ with reference distribution $P_\xi = N(0, I_M)$, where $f_\theta(x, \xi) = \langle x, \mu + \boldsymbol{A}\xi \rangle$. In this case $f_\theta$ follows a Gaussian distribution with mean $\mu$ and covariance $\boldsymbol{A}\boldsymbol{A}^\top$. Ensemble of $M$ neural networks $g_{\theta_1}, \ldots, g_{\theta_M}$ is also a special case of hypermodel with parameter $\theta = (\theta_1, \ldots, \theta_M) \in \mathbb{R}^{d \times M}$ with reference distribution $P_\xi = \mathcal{U}(e_1, \ldots, e_M)$ being uniform distribution over one-hot vectors, where $f_\theta(x, \xi) = g_{\langle \theta, \xi \rangle}(x)$. In general, the hypermodel $f_\theta(\cdot)$ can be any function, e.g. neural networks, transforming the reference distribution $P_\xi$ to arbitrary distribution. We adopt a class of hypermodel that can be represented as an additive function

$$\underbrace{f_\theta(x, \xi)}_{\text{``Posterior'' Hypermodel}} = \underbrace{f_\theta^L(x, \xi)}_{\text{Learnable function}} + \underbrace{f^P(x, \xi)}_{\text{Fixed prior model}} \tag{2}$$

The prior model $f^P$ represents the prior bias and uncertainty and has NO trainable parameters. The learnable function is initialized to output value near zero and is then trained by fitting the data. The resultant sum $f_\theta$ produces reasonable predictions for all probable values of $\xi$. Variations of a prediction $f_\theta(x, \cdot)$ as a function of $\xi$ indicate the epistemic uncertainty estimation. The prior model $f^P(\cdot, \xi)$ can be viewed as a prior distribution of the true model $f^*$, which is the true function that generates the data. The hypermodel $f_\theta(\cdot, \xi)$ can be viewed as a trained approximate posterior distribution of the true model $f^*$ given the data. Similar decomposition in Equation (2) is also used in (Dwaracherla et al., 2020; Osband et al., 2021; Li et al., 2022a). We will discuss the implementation details and clarify the importance of difference between our work and prior works in Appendix A.

## 3 ALGORITHM

We now develop a novel DQN-type algorithm for large-scale RL problems with value function approximation, called `HyperFQI`. `HyperFQI` uses a hypermodel to maintain a probability distribution over the action-value function and aims to approximate the posterior distribution of $Q^* := Q_M^{\pi^*}$. The hypermodel in this context is a function $f_\theta : \mathcal{S} \times \mathcal{A} \times \Xi \to \mathbb{R}$ parameterized by $\theta \in \Theta$ and $\Xi$ is the index space. Hypermodel is then trained by minimizing the loss function motivated by fitted Q-iteration (FQI), a classical method (Ernst et al., 2005) for value function approximation. `HyperFQI` selects the action based on sampling indices from reference distribution $P_\xi$ and then taking the action with the highest value from hypermodels applying these indices. This can be viewed as an value-based approximate Thompson sampling via Hypermodel.

Alongside the learning process, `HyperFQI` maintains two hypermodels, one for the current value function $f_\theta$ and the other for the target value function $f_{\theta^-}$. `HyperFQI` also maintains a buffer of transitions $D = \{(s, a, r, s', \mathbf{z})\}$, where $\mathbf{z} \in \mathbb{R}^M$ is the algorithm-generated perturbation random vector sampled from the perturbation distribution $P_\mathbf{z}$. For a transition tuple $d = (s, a, r, s', \mathbf{z}) \in D$ and given index $\xi$, the temporal difference (TD) error for hypermodel is

$$\ell^{\gamma, \sigma}(\theta; \theta^-, \boldsymbol{\xi}^-, \xi, d) = \left( f_\theta(s, a, \xi) - (r + \sigma \xi^\top \mathbf{z} + \gamma \max_{a' \in \mathcal{A}} f_{\theta^-}(s', a', \boldsymbol{\xi}^-(s'))) \right)^2 \tag{3}$$

where $\theta^-$ is the target parameters, and the $\sigma$ is a hyperparameter to control the injected noise by algorithm. $\boldsymbol{\xi}^-$ is the target index mapping such that $\boldsymbol{\xi}^-(s)$ one-to-one maps each state $s \in \mathcal{S}$ to a random vector from $P_\xi$, all of which are independent with $\xi$.[1] The algorithm update the hypermodel for value function by minimizing

$$L^{\gamma, \sigma, \beta}(\theta; \theta^-, \boldsymbol{\xi}^-, D) = \mathbb{E}_{\xi \sim P_\xi} [\sum_{d \in D} \frac{1}{|D|} \ell_{z^-}^{\gamma, \sigma}(\theta; \theta^-, \boldsymbol{\xi}^-(s'), \xi, d)] + \frac{\beta}{|D|} \|\theta\|^2 \tag{4}$$

where $\beta \geq 0$ is the prior regularization parameter. Note the target hypermodel is necessary for stabilizing the optimization and reinforcement learning process, as discussed in target Q-network literature (Mnih et al., 2015; Li et al., 2022b). We optimize the loss function Equation (4) using stochastic gradient descent (SGD) with a mini-batch of data $\tilde{D}$ and a batch of indices $\tilde{\Xi}$ from $P_\xi$.

---

[1]To clarify, the random vector $\boldsymbol{\xi}^-(s)$ remains the same vector if we do not resample the mapping $\boldsymbol{\xi}^-$.

That is, we take gradient descent with respect to the sampled version of loss

$$\tilde{L}(\theta; \theta^-, \boldsymbol{\xi}^-, \tilde{D}) = \frac{1}{|\tilde{\Xi}|} \sum_{\xi \in \tilde{\Xi}} \left( \sum_{d \in \tilde{D}} \frac{1}{|\tilde{D}|} \ell^{\gamma, \sigma}(\theta; \theta^-, \boldsymbol{\xi}^-, \xi, d) \right) + \frac{\beta}{|D|} \|\theta\|^2 \tag{5}$$

We summarize the `HyperFQI` algorithm: At each episode $k$, `HyperFQI` samples an index mapping $\boldsymbol{\xi}_k$ from the index distribution $P_\xi$ and then take action by maximizing the associated hypermodel $f_\theta(\cdot, a, \boldsymbol{\xi}_k(\cdot))$, which we call index sampling (IS) action selection.[2] This can be viewed as an value-based approximate Thompson sampling. The algorithm updates the hypermodel parameters $\theta$ in each episode according to Equation (5), and updates the target hypermodel parameters $\theta^-$ periodically. The algorithm also maintains a replay buffer of transitions $D$, which is used to sample a mini-batch of data $\tilde{D}$ for training the hypermodel.

---

**Algorithm 1** HyperFQI for RL

---

1: **Input:** Initial parameter $\theta_{\text{init}}$, Hypermodel for value $f_\theta(s = \cdot, a = \cdot, \boldsymbol{\xi} = \cdot)$ with dist. $P_\xi$.
2: Initialize $\theta = \theta^- = \theta_{\text{init}}$, train step $j = 0$ and buffer $D$
3: **for** each episode $k = 1, 2, \ldots$ **do**
4:     **Sample index** mapping $\boldsymbol{\xi}_k \sim P_\xi$
5:     Set $t = 0$ and **Observe** $S_{k,0} \sim \rho$
6:     **repeat**
7:         **Select** $A_{k,t} = \arg\max_{a \in \mathcal{A}} f_\theta(S_{k,t}, a, \boldsymbol{\xi}_k(S_{k,t}))$
8:         **Observe** $R_{k,t+1}$ and $S_{k,t+1}$ **from environment**
9:         **Sample** $\mathbf{z}_{k,t+1} \sim P_\mathbf{z}$ and $D.\mathbf{add}((S_{k,t}, A_{k,t}, R_{k,t+1}, S_{k,t+1}, \mathbf{z}_{k,t+1}))$
10:         Increment step counter $t \leftarrow t + 1$
11:         $\theta, \theta^-, j \leftarrow \mathbf{update}(D, \theta, \theta^-, \boldsymbol{\xi}^- = \boldsymbol{\xi}_k, t, j)$
12:     **until** $S_t = s_{\text{terminal}}$
13: **end for**

---

This algorithm offers several advantages over existing methods. First, it is computationally efficient due to the nature of incremental update and scalable to large-scale problems. Second, it is compatible with existing deep RL algorithms and can be used as a drop-in replacement for the Q-network in DQN-type methods. Finally, it is easy to implement and can be applied to a wide range of problems.

## 4 EXPERIMENTAL STUDIES

This section evaluates the efficiency and scalability of our HyperFQI. Our experiments on the DeepSea demonstrate its high data and computation efficiency, achieving polynomial performance. We also showcase the scalability of our approach by successfully processing large-scale states with a size of $100^2$. In addition, we evaluate the scalability using the Atari games, where our Hyper-FQI performs exceptionally well in processing states with pixels. Furthermore, our approach can achieve human-level performance with remarkable data and computation efficiency in Atari games. **To highlight**, HyperFQI is (1) the first algorithm in the literature solving DeepSea at a large scale up to $100^2$ states, and (2) also the first one achieving human-level performance in Atari games, considering both data and computation efficiency.

### 4.1 COMPUTATIONAL RESULTS FOR DEEP EXPLORATION

We demonstrate the exploration effectiveness of our HyperFQI using DeepSea, a reward-sparse environment that demands deep exploration. DeepSea offers only two actions: moving left or right; see Appendix B. The agent receives a reward of 0 for moving left, and a penalty of $-(0.01/N)$ for moving right, where $N$ denotes the size of DeepSea. The agent earns a reward of 1 upon reaching the lower-right corner of the DeepSea, making it optimal for the agent to continuously move towards the right and get the total return 0.99. We repeat all the experiments with different 10 seeds on DeepSea.

---

[2]To clarify, the random vector $\boldsymbol{\xi}_k(s)$ remains the same vector within the episode $k$.

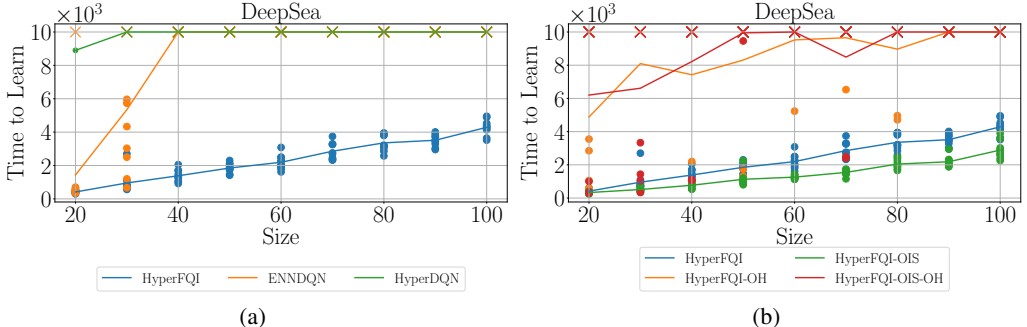

(a)                                                                         (b)

Figure 2: Experimental results on DeepSea. The y-axis represents the number of episodes required to learn the optimal policy for a specific problem size. The symbol × indicates that the algorithm was unable to solve the problem within $10e3$ episodes. (a) The performance of various baselines. We have not included the performance of DoubleDQN (Van Hasselt et al., 2016) and BootDQN (Osband et al., 2018), as both algorithms were unable to solve DeepSea-20 within $10e3$ episodes (see Appendix C.1 for detailed results). (b) The performance of different variants with our HyperFQI.

**Baselines Result**: We define the $Time\ to\ Learn(N) := mean\{K | \bar{R}_K \geq 0.99\}$, which serves as an evaluation metric for algorithm performance on DeepSea with size $N$. The $\bar{R}_K$ represents the total return obtained by the agent after $K$ episodes of interaction, and we assess $\bar{R}_K$ 100 times. Overall, the $Time\ to\ Learn(N)$ indicates the number of episodes needed to learn the optimal policy. As shown in Figure 2(a), our HyperFQI can achieve superior performance compared to other baselines. Based on the structure of DeepSea, we can deduce that discovering the optimal policy requires at least $N^2/2$ episodes, as all accessible states must be traversed. Our experiments demonstrate that our HyperFQI can achieve the optimal policy within the aforementioned episodes range, which fully demonstrates the data efficiency of our algorithm.

Notably, HyperDQN (Li et al., 2022a), which has demonstrated effective deep exploration, only can learn the optimal policy for a DeepSea-20 within $10e3$ episodes. These results provide evidence for the effectiveness of our network structure and suggest that the update method used in HyperFQI enhances the model's ability to capture uncertainty and promote effective exploration. Our approach samples multiple indexes for each state rather than each batch, leading to more accurate expectation estimation compared to HyperDQN. Additionally, our method has a more efficient network initialization than HyperDQN. For a detailed comparison, please refer to the Appendix A.1.3.

The ENNDQN, which was adapted from Osband et al. (2023), struggles to solve DeepSea as its size increases. Compared to our approach, ENNDQN includes the original input as a component of the final ENN layer's input. Both HyperFQI and ENNDQN share the same feature network, and the parameters in our output layer (hypermodel) remain constant when scaling up the problem. However, the ENN layer requires a greater number of parameters and computational workload, especially as the problem's scale increases. In the case of DeepSea-20, the number of parameters in the ENN layer is almost 20 times larger than our hypermodel. These findings demonstrate that the network architecture of our HyperFQI can enhance both computational efficiency and scalability when dealing with large scale problem.

**Variants of HyperFQI**: We can produce various variants based on the framework of HyperFQI, including HyperFQI-OIS, which employs optimistic index sampling (OIS) action selection (refer to Appendix A.1.2 for details). Furthermore, we substitute the Gaussian distributional index with a one-hot index under two different action selections, resulting in HyperFQI-OH and HyperFQI-OIS-OH. In Figure 2(b), we compare the performance of different variants of our approach. Our HyperFQI-OIS impressively outperforms HyperFQI by utilizing optimistic index sampling action selection to achieve more optimistic estimates, which can enhance exploration. The OIS method does not increase much computation as we set the dimension $M$ to 4. We observed that HyperFQI-OH is not effective in DeepSea as the Gaussian distributional index provides superior expectation estimation compared to the one-hot index. However, subsequent experiments show that increasing the dimension of the one-hot index can improve exploration.

**Ablation Study**: We consider how the dimension $M$ of the index affects our methods. Figure 3 demonstrates that increasing the $M$ of the one-hot index can lead to improved estimation of expectations, which in turn can enhance exploration. HyperFQI-OIS also can result in better performance when using the one-hot index with $M = 16$, which is shown in Appendix C.1. On the other hand, increasing the dimension of the Gaussian distributional index can actually hurt the algorithm's performance because it becomes more difficult to estimate the expectation in Equation (5) under $P_\xi$ higher index dimension. However, there are ways to mitigate this problem. For a given dimension $M$, increasing the number of indices $|\tilde{\Xi}|$ of Equation (5) during the update phase can result in more accurate estimates, as demonstrated in Appendix 6. However, this comes at the cost of increased computation, which slows down the algorithm. To strike a balance between performance and computation, we have chosen $M = 4$ and $|\tilde{\Xi}| = 20$ as our default hyper-parameters. In addition, we have also investigated the effect of other hyper-parameters on our methods, as shown in Appendix C.1.

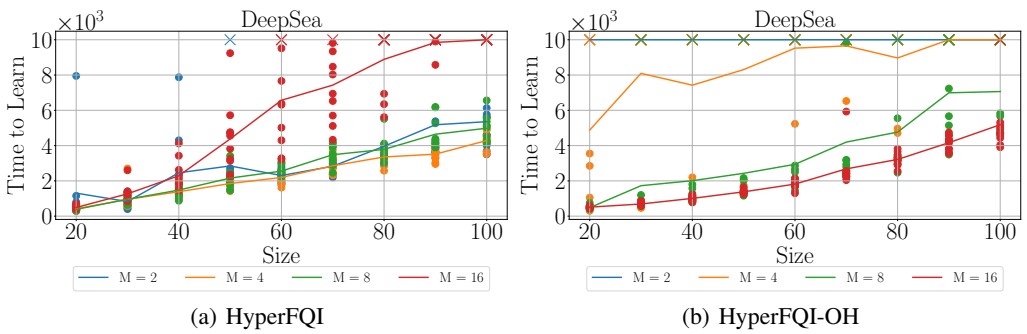

(a) HyperFQI            (b) HyperFQI-OH

Figure 3: Ablation results under different index dimension $M$.

## 4.2 ATARI RESULTS

We assess the computational complexity of various methods on the Arcade Learning Environment (Bellemare et al., 2013) using IQM (Agarwal et al., 2021) as the evaluation criterion. An IQM score of 1 indicates that the algorithm's performance is comparable to that of a human. We examine our HyperFQI with six baselines: DQN (Nature) (Van Hasselt et al., 2016), Rainbow (Hessel et al., 2018), DER (Van Hasselt et al., 2019), HyperDQN (Li et al., 2022a), BBF (Schwarzer et al., 2023) and EfficientZero (Ye et al., 2021). Specially, the EfficientZero is a state-of-the-art model-based method, while the others are value-based methods.

Figure 1 illustrates the relationship between model parameters and the amount of training data required to achieve human-level performance. Our HyperFQI achieves human-level performance with minimal parameters and relatively little training data, outperforming other methods. Notably, the Convolutional layers in our model are the same as those in Rainbow and account for only a small fraction (about 13%) of the overall model parameters. This suggests that the Fully Connected layers dominate the computational complexity of the model, and as we know, the computational complexity of a Fully Connected layer is directly proportional to the number of parameters. In fact, our model employs the first Fully Connected layer with just 256 units, which is even fewer than DQN (Natural). Consequently, our HyperFQI offers superior computational performance due to having fewer parameters in the Fully Connected layer than other baselines.

We assessed various variants of our HyperFQI on 26 Atari games using 2 million training data, and we presented their performance in Table 1. In addition, we implemented our version of DDQN, named DDQN (ours), using the same hyper-parameters and network initialization scheme as our HyperFQI. In comparison, We report the results of vanilla DDQN[†] from Hessel et al. (2018). Our results show that DDQN (ours) outperforms DDQN[†] due to the increased data efficiency provided by our hyper-parameters and network initialization. Furthermore, our HyperFQI demonstrates superior performance compared to DDQN, as HyperFQI includes an additional hypermodel that enables deep exploration in Atari games. Additional, our findings indicate that all methods performed similarly, implying that the OIS method or one-hot vector do not generate significant differences in complex networks such as Convolutional layers. More detailed results for each Atari game are available

|  | IQM | Median | Mean |
|---|---|---|---|
| DDQN[†] | 0.13 (0.11, 0.15) | 0.12 (0.07, 0.14) | 0.49 (0.43, 0.55) |
| DDQN (ours) | 0.70 (0.69, 0.71) | 0.55 (0.54, 0.58) | 0.97 (0.95, 1.00) |
| HyperFQI | 1.22 (1.15, 1.30) | 1.07 (1.03, 1.14) | 1.97 (1.89, 2.07) |
| HyperFQI-OH | 1.28 (1.21, 1.35) | 1.13 (1.10, 1.18) | 2.03 (1.93, 2.15) |
| HyperFQI-OIS | 1.15 (1.09, 1.22) | 1.12 (1.02, 1.18) | 2.02 (1.91, 2.16) |
| HyperFQI-OIS-OH | 1.25 (1.18, 1.32) | 1.10 (1.04, 1.17) | 2.02 (1.93, 2.12) |

Table 1: Performance profiles of our HyperFQI with different variant.

in Appendix C.2, where we visualize the relative improvement compared to other baselines and the learning curve of our variants. The results demonstrate the better exploration efficiency of our HyperFQI than Rainbow and the robustness of all our variants across all Atari games.

## 5 ANALYSIS

In this section, we try to explain the intuition behind the HyperFQI algorithm and how it achieves efficient deep exploration. We also provide a regret bound for HyperFQI in finite horizon time-inhomoegeneous MDPs. First, we describe the HyperFQI algorithm in Algorithm 1 when specified to tabular problems.

**Tabular HyperFQI.** Let $f_\theta(s, a, \xi) = \mu_{sa} + m_{sa}^\top \xi + \mu_{0,sa} + \sigma_0 \mathbf{z}_{0,sa}^\top \xi$ where $\theta = (\mu \in \mathbb{R}^{SA}, m \in \mathbb{R}^{SA \times M})$ are the parameters to be learned, and $\mathbf{z}_{0,sa} \in \mathbb{R}^M$ is a random vector from $P_{\mathbf{z}}$ and $\mu_{0,sa}, \sigma_0$ is a prior mean and prior variance for each $(s, a)$. The regularizer in Equation (4) becomes $\beta \|\theta\|^2 = \beta \sum_{s,a} (\mu_{sa}^2 + \|m_{sa}\|^2)$. Let the set $E_{k,sa}$ record the time index the agent encountered $(s, a)$ in the $k$-th episode $E_{k,sa} = \{t : (S_{k,t}, A_{k,t}) = (s, a)\}$. Let $N_{k,sa} = \sum_{\ell=1}^{k-1} \sum_{t=0}^{H-1} \mathbb{1}_{(S_{\ell,t}, A_{\ell,t})=(s,a)}$ denoting the counts of visitation for state-action pair $(s, a)$ prior to episode $k$.

**Closed-form incremental update.** Let $\beta = \sigma^2/\sigma_0^2$. Then, given the dataset $D = H_k$ and target noise $\xi^-$ before episode $k$, HyperFQI in Algorithm 1 with $\gamma = 1$ would yield the following closed-form backward iterative procedure $\theta_k^{(t+1)} = (\mu_k^{(t+1)}, m_k) \to \theta_k^{(t)} = (\mu_k^{(t)}, m_k)$ for all $t = H - 1, H - 2, \ldots, 0$,

$$m_{k,sa} = \frac{\sigma \sum_{\ell=1}^{k-1} \sum_{t \in E_{\ell,sa}} \mathbf{z}_{\ell,t+1} + \beta \sigma_0 \mathbf{z}_{0,sa}}{N_{k,sa} + \beta}, \forall (s, a) \in \mathcal{S} \times \mathcal{A} \tag{6}$$

$$\mu_{k,sa}^{(t)} = \frac{\sum_{\ell=1}^{k-1} \sum_{t \in E_{\ell,sa}} y_{k,\xi^-}^{(t+1)} + \beta \mu_{0,sa}}{N_{k,sa} + \beta}, \forall (s, a) \in \mathcal{S} \times \mathcal{A} \tag{7}$$

where $y_{k,\xi^-}^{(t+1)} = R_{k,t+1} + \max_{a' \in \mathcal{A}} f_{\theta_k^{(t+1)}}(S_{k,t+1}, a', \xi^-)$.

### 5.1 HOW DOES HYPERFQI DRIVES EFFICIENT DEEP EXPLORATION?

In this section, we highlight the key components of HyperFQI that enable efficient deep exploration. We consider a simple example (adapted from (Osband et al., 2019b)) to understand the HyperFQI's learning rule in Equations (4) and (5) and the role of hypermodel, and how they together drive efficient deep exploration.

**Example 5.1.** Consider a fixed horizon MDP $\mathcal{M}$ with four states $\mathcal{S} = \{1, 2, 3, 4\}$, two actions $\mathcal{A} = \{up, down\}$ and a horizon of $H = 6$. Let $\mathcal{H}$ be the list of all transitions observed so far, and let $\mathcal{H}_{s,a} = ((\hat{s}, \hat{a}, r, s') \in \mathcal{H} : (\hat{s}, \hat{a}) = (s, a))$ contain the transitions from state-action pair $(s, a)$. Suppose $|\mathcal{H}_{4,down}| = 1$, while for every other pair $(s, a) \neq (4, down), |\mathcal{D}_{s,a}|$ is very large, virtually infinite. Hence, we are highly certain about the expected immediate rewards and transition probabilities except for $(4, down)$. Assume that this is the case for all time periods $t \in \{0, 1, \ldots, 5\}$.

HyperFQI produces a sequence of action-value functions $f_0, f_1, \ldots f_5$. In Figure 4, each triangle in row $s$ and column $t$ contains two smaller triangles that are associated with action-values of $up$ and $down$ actions at state $s$. The shade on the smaller triangle shows the uncertainty estimates in the $f_t(s, a, \xi)$, specifically the variance $\text{Var}_\xi (f_t(s, a, \xi))$. The dotted lines show plausible transitions, except at $(4, down)$. Since we are uncertain about $(4, down)$, any transition is plausible. We will

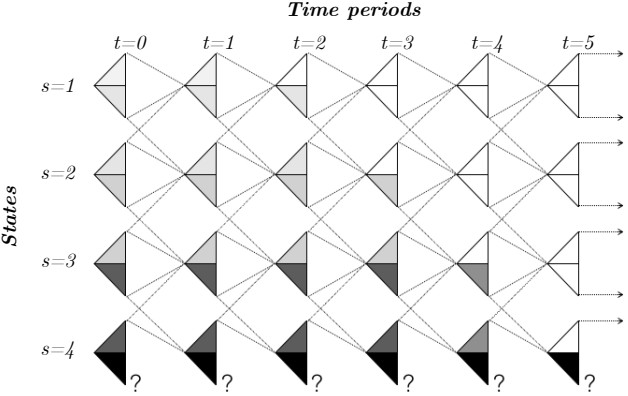

Figure 4: Example to illustrate how HyperFQI achieves deep exploration. We can see the propagation of uncertainty from later time period to earlier time period in the figure. Darker shade indicates higher degree of uncertainty.

show how HyperFQI efficiently computes the uncertainty propagation backward in time, which can be visualized as progressing leftward in Figure 4. Also, we will show the ability to estimate the degree of uncertainty drives deep exploration. This can be explained by the incremental closed-form update in tabular setting described in Equations (6) and (7). A key property is that, with logarithmically small additional dimension $M$, hypermodel can approximate the posterior distribution of the optimal $Q^*$-values with low computation cost. This is formalized in the following.

**Lemma 5.2** (Approximate posterior variance)**.** *For $m_k$ defined in Equation (6) with $\mathbf{z} \sim$ Uniform$(\mathbb{S}^{M-1})$. For any $k \geq 1$, a good event $\mathcal{G}_k(s, a)$ is defined as*

$$\mathcal{G}_k(s, a) = \left\{ \|m_k(s, a)\|^2 \in \left( \frac{\sigma^2}{N_{k,sa} + \beta}, \frac{3\sigma^2}{N_{k,sa} + \beta} \right) \right\}.$$

*Then the joint event $\cap_{(s,a,k) \in \mathcal{S} \times \mathcal{A} \times [K]} \mathcal{G}_k(s, a)$ holds w.p. at least $1 - \delta$ if $M \simeq \log(SAK/\delta)$.*

## 5.2 REGRET BOUND

Denote the regret of a policy $\pi_k$ over episode $k$ by $\Delta_k := \mathbb{E}_{M, \text{alg}}[V_M^{\pi^*}(s_{k,0}) - V_M^{\pi_k}(s_{k,0})]$, where $\pi^*$ is an optimal policy for $M$. The goal of the agent is equivalent to minimizing the expected total regret up to episode $K$, $\text{Regret}(K, \text{alg}) := \mathbb{E}_{\text{alg}} \sum_{k=1}^{K} \Delta_k$, where the subscript $\text{alg}$ under the expectation indicates that policies are generated through algorithm $\text{alg}$. Note that the expectation in Equation (12) is over the random transitions and rewards, the possible randomization in the learning algorithm $\text{alg}$, and also the unknown MDP $M$ based on the agent designer's prior beliefs. Finally, we show that, with the help of hypermodel approximation property in Lemma 5.2, HyperFQI achieves efficient deep exploration in finite horizon time-inhomogeneous MDPs. This is formalized in the following theorem.

**Theorem 5.3** (Regret bound of HyperFQI)**.** *Consider an HyperFQI with an infinite buffer, greedy actions and with tabular representation. Under Assumptions E.1 and E.2 with $\beta \geq 3$, if the tabular HyperFQI is applied with planning horizon $H$, and parameters with $(M, \mu_0, \sigma, \sigma_0)$ satisfying $M \simeq \log(|\mathcal{X}||\mathcal{A}|HK)$, $(\sigma^2/\sigma_0^2) = \beta$, $\sigma \geq \sqrt{3}H$ and $\mu_{0,s,a} = H$, then for all $K \in \mathbb{N}$,*

$$\text{Regret}(K, \textit{HyperFQI}) \leq 18H^2 \sqrt{\beta |\mathcal{X}||\mathcal{A}|K \log_+(1 + |\mathcal{X}||\mathcal{A}|HK)} \log_+ \left( 1 + \frac{K}{|\mathcal{X}||\mathcal{A}|} \right), \quad (8)$$

*where $\log_+(x) = \max\{1, \log(x)\}$.*

*Remark* 5.4. The Assumption E.1 is common in the literature of regret analysis, e.g. (Osband et al., 2019b; Jin et al., 2018). The relationship between two set $\mathcal{S}$ and $\mathcal{X}$ is described in Assumption E.1. The Assumption E.2 is common in the Bayesian regret literature (Osband et al., 2013; 2019b; Osband & Van Roy, 2017; Lu & Van Roy, 2019). Our regret bound $\mathcal{O}(H^2\sqrt{|\mathcal{X}||\mathcal{A}|K})$ matches the best known Bayesian regret bound in the literature, say RLSVI (Osband et al., 2019b) and PSRL (Osband & Van Roy, 2017) which while our HyperFQI provide the computation and scalability benefit that RLSVI and PSRL do not have. We believe the `HyperFQI` algorithm provides a bridge for the theory and practice in RL.

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

# A    HyperFQI ALGORITHM DETAILS

In this section, we describe more details of the proposed HyperFQI. First, we describe the general treatment for the **update** function (in line 11 of HyperFQI) in the following Algorithm 2. Then, in Appendix A.1, we provide the implementation details of HyperFQI with deep neural network (DNN) function approximation. We want to emphasize that all experiments done in this article is using **Option 1** with DNN value function approximation. In Appendix A.2, we describe the closed-form update rule (**Option 2**) when the tabular representation of the value function is exploited. Note that the tabular version of HyperFQI is only for the clarity of analysis and understanding.

---

**Algorithm 2 update**

---

1: **Input:** buffer $D$, $\theta, \theta^-, \boldsymbol{\xi}^-$, agent step $t$, train step $j$
2: **if** $t \mod \text{training\_freq} = 0$ **then**
3:     **repeat**
4:         Obtain $\theta$ by optimizing the loss $L^{\gamma,\sigma,\beta}(\theta; \theta^-, \boldsymbol{\xi}^-, D)$ in Equation (4):
           – **Option** (1) with gradient descent w.r.t. the mini-batch sampled loss Equation (5);
           – **Option** (2) with closed-form solution in Equations (6) and (7).
5:         Increment $j \leftarrow j + 1$
6:         **if** $j \mod \text{target\_update\_freq} = 0$ **then**
7:            $\theta^- \leftarrow \theta$
8:         **end if**
9:     **until** $j \mod \text{sample\_update\_ratio} \times \text{training\_freq} = 0$
10: **end if**
11: **Return:** $\theta, \theta^-, j$.

---

## A.1    FUNCTION APPROXIMATION WITH DEEP NEURAL NETWORKS

Here we describe the implementation details of HyperFQI with deep neural networks and the main difference compared to baselines.

### A.1.1    NETWORK ARCHITECTURE OF HYPERFQI

In our implementation, we only apply hypermodel to the output layer of our network, which will not result in much parameters and provide better expectation estimate. Suppose the hidden layers in neural networks forms the nonlinear feature mapping $\phi_w(\cdot)$ with parameters $w$. Our hypermodel takes the random index $\xi \in \mathbb{R}^M$ from reference distribution $P_\xi$ and outputs the weights for output layer. It's worth to note that our hypermodel only outputs the weights but not bias for output layer, which is indicated by following equation with trainable $\theta = \{\boldsymbol{A}, b, w\}$ and fixed parameters $\{\boldsymbol{A}_0, b_0, w_0\}$

$$
\begin{aligned}
f_\nu(x, \xi) &= \underbrace{\langle \boldsymbol{A}\xi + b, \phi_w(x) \rangle}_{\text{\textbf{Learnable} } f_\nu^L(x,\xi)} + \underbrace{\langle \boldsymbol{A}_0\xi + b_0, \phi_{w_0}(x) \rangle}_{\text{\textbf{Fixed prior} } f^P(x,\xi)} \\
&= \underbrace{\langle \boldsymbol{A}\xi, \phi_w(x) \rangle}_{\sigma_\theta^L(x,\xi)} + \underbrace{\langle \boldsymbol{A}_0\xi, \phi_{w_0}(x) \rangle}_{\sigma^P(x,\xi)} + \underbrace{\langle b, \phi_w(x) \rangle}_{\mu_\theta^L(x)} + \underbrace{\langle b_0, \phi_{w_0}(x) \rangle}_{\mu^P(x)}.
\end{aligned} \tag{9}
$$

Through our formulation ( 9), HyperFQI can accurately estimate the learnable mean $\mu_\theta^L(x)$, which relies solely on the original input $x$, and the variation prediction $\sigma_\theta^L(x, \xi)$, which is dependent on both the original input $x$ and random index $\xi$. This allows our hypermodel to capture uncertainty better, without being influenced by other components that may only depend on the random index $\xi$ like HyperDQN (Li et al., 2022a). The fixed prior model also offers prior bias and prior variation through the functions $\mu^P(x)$ and $\sigma^P(x, \xi)$. This priormodel is NOT trainable so that it will not bring much computation, and designed to provide better exploration in the early stage of training. We use Xavier Normal method to initialize our entire network except the part of priormodel which is corresponding to hypermodel. For the initialization of prior model, we follow the method described in Dwaracherla et al. (2020). In this way, each row of priormodel is sampled from the unit hypersphere, which guarantees that the output of priormodel can follow a desired Gaussian distribution.

### A.1.2 TRAINING DETAILS FOR HYPERFQI

For better expectation estimate, we sample multiple random indexes for each state and compute the empirical expectation in update stage. We let $|\tilde{\Xi}|$ denote the number of random indexes for each state. For continuous or uncountable infinite state space, we just sample independent $\xi^-$ for each data tuple $d$ in the mini-batch.

We tuned the some hyper-parameters on our HyperFQI and listed them in Table 2, and other hyper-parameters for Atari games are the same as Rainbow Hessel et al. (2018). Notice that we use a single configuration for all Atari games we test. Also we use a single configuration for the deepsea environments with variant sizes.

| Hyper-parameters | Atari Setting | DeepSea Setting |
|---|---:|---:|
| discount factor $\gamma$ | 0.99 | 0.99 |
| learning rate | 0.001 | 0.001 |
| minibatch size $|\tilde{D}|$ | 32 | 128 |
| index dim $M$ | 4 | 4 |
| # Indices $|\tilde{\Xi}|$ for approximation | 20 | 20 |
| $n$-step target | 5 | 1 |
| target network freq | 5 | 4 |
| sample update ratio | 1 | 1 |
| training freq | 1 | 1 |
| hidden units | 256 | 64 |
| min replay size for sampling | 2,000 steps | 128 steps |
| memory size | 500,000 steps | 1000000 steps |

Table 2: Hyper-parameters of our HyperFQI

As per the framework of HyperFQI, we can generate various variants. By default, a Gaussian distribution is used for $P_\xi$, but this can be changed to a one-hot index, referred to as HyperFQI-OH.

The action selection and computation of $Q$-target can also be modified by sampling multiple indexes and computing multiple $Q$-values for each action under one exact state. The optimal action is then selected based on these multiple $Q$-values. This variant is called HyperFQI-OIS and is described in the following.

**Optimistic Index Sampling.** To make agent's behavior more optimistic, in each episode $k$, we can sample $N_{\text{OIS}}$ indices $\xi_{k,1}, \ldots, \xi_{k,N_{\text{OIS}}}$ and take action by maximizing the associated hyper-model $f_k(\cdot, a) = \max_{n \in [N_{\text{OIS}}]} f_\theta(\cdot, a, \xi_{k,n})$, which we call optimistic index sampling (OIS) action selection.

In the hypermodel training part, for any transition tuple $d = (s, a, r, s', \mathbf{z})$, we also sample multiple indices $\xi_1^-, \ldots, \xi_{N_{\text{OIS}}}^-$ and modify the target computation in Equation (3) as

$$r + \sigma \xi^\top \mathbf{z} + \gamma \max_{a' \in \mathcal{A}} \max_{n \in [N_{\text{OIS}}]} f_{\theta^-}(s', a', \xi_n^-).$$

This modification in target computation boosts uncertainty propagation from later states to earlier states, which is beneficial for deep exploration. We call this variant **HyperFQI-OIS**.

### A.1.3 DIFFERENCE COMPARED TO PRIOR WORKS

There are several methods which have similar structure as our HyperFQI but fail to achieve good performance as ours.

HyperModel (Dwaracherla et al., 2020) employs hypermodels to represent epistemic uncertainty and facilitate exploration. However, the use of hypermodels over entire networks leads to an extensive number of parameters and optimization challenges. As a result, applying HyperModel (Dwaracherla et al., 2020) to address large-scale problems such as Atari games can be extremely difficult. Our

HyperFQI only apply hypermodel to the output layer, which provide better exploration and efficient computation in large scale problem.

HyperDQN (Li et al., 2022a) shares a similar structure with our HyperFQI and has shown promising results in exploration. Nevertheless, it falls short in handling the DeepSea environment, which demands deep exploration. We have improved HyperDQN by simplifying our hypermodel, as shown in equation (9). Our HyperFQI estimates the mean $\mu$ solely based on the original input $x$, and estimates the variation $\sigma$ based on both the original input $x$ and a random index $\xi$. However, HyperDQN use hypermodel to generate the both weights and bias for output layer, resulting in some redundant components, such as functions that depend solely on the random index $\xi$ or functions that depend only on the parameters of the hypermodel. These components lack a clear semantic explanation, rendering HyperFQI unsuitable for estimating uncertainty. Additionally, we apply $NpS$ indexes to each transition in update stage to improve the expectation estimate, whereas HyperDQN only applies them to batch transitions. We have found that initializing the hypermodel with Xavier Normal can improve optimization. The combination of these factors leads to our HyperFQI outperforming HyperDQN on both DeepSea and Atari, as demonstrated in Section 4.

The ENN approach, as described in Osband et al. (2023), shows promise for capturing epistemic uncertainty and has demonstrated efficiency on various tasks. We have implemented the ENNDQN by their description on bsuite (Osband et al., 2019a). Except for the update method and network structure, other settings are the same as our HyperFQI. In the update stage, they use "stop gradient" between feature layers and final ENN layers. For the network structure, they concatenate the original input $x$, feature $\phi(x)$ and random index $\xi$ as the input for the ENN layer, and use ensemble priormodel for ENN layer but don't have priormodel for feature network. The network structure leads to larger parameters when processing tasks at a large scale, causing significant computation and optimization challenges. As shown in Section 4.1, ENNDQN performs well on DeepSea-20 but struggles with larger scale of the problem. This difficulty arises because the input $x$ in DeepSea has a dimension of $N^2$, which is too large for the ENN layer to handle. We designed our HyperFQI to take only a random index $\xi$ as input, resulting in a more efficient computation with fewer parameters.

## A.2 TABULAR SETTING

In this section, we try to explain the intuition behind the HyperFQI algorithm and how it achieves efficient deep exploration. We also provide a regret bound for HyperFQI in finite horizon time-inhomoegeneous MDPs in Appendix E. First, we describe the HyperFQI algorithm in Algorithm 1 when specified to tabular problems.

**Tabular HyperFQI.** Let $f_\theta(s, a, \xi) = \mu_{sa} + m_{sa}^\top \xi + \mu_{0,sa} + \sigma_0 \mathbf{z}_{0,sa}^\top \xi$ where $\theta = (\mu \in \mathbb{R}^{SA}, m \in \mathbb{R}^{SA \times M})$ are the parameters to be learned, and $\mathbf{z}_{0,sa} \in \mathbb{R}^M$ is a random vector from $P_{\mathbf{z}}$ and $\mu_{0,sa}, \sigma_0$ is a prior mean and prior variance for each $(s, a)$. The regularizer in Equation (4) becomes $\beta \|\theta\|^2 = \beta \sum_{s,a} \left( \mu_{sa}^2 + \|m_{sa}\|^2 \right)$. Let the set $E_{k,sa}$ record the time index the agent encountered $(s, a)$ in the $k$-th episode $E_{k,sa} = \{t : (S_{k,t}, A_{k,t}) = (s, a)\}$. Let $N_{k,sa} = \sum_{\ell=1}^{k-1} \sum_{t=0}^{H-1} \mathbb{1}_{(S_{\ell,t}, A_{\ell,t})=(s,a)}$ denoting the counts of visitation for state-action pair $(s, a)$ prior to episode $k$.

**Closed-form incremental update.** Let $\beta = \sigma^2 / \sigma_0^2$. Then, given the dataset $D = H_k$ and target noise $\xi^-$ before episode $k$, HyperFQI in Algorithm 1 with $\gamma = 1$ would yield the following closed-form backward iterative procedure $\theta_k^{(t+1)} = (\mu_k^{(t+1)}, m_k) \rightarrow \theta_k^{(t)} = (\mu_k^{(t)}, m_k)$ for all $t = H - 1, H - 2, \ldots, 0$,

$$m_{k,sa} = \frac{\sigma \sum_{\ell=1}^{k-1} \sum_{t \in E_{\ell,sa}} \mathbf{z}_{\ell,t+1} + \beta \sigma_0 \mathbf{z}_{0,sa}}{N_{k,sa} + \beta}, \forall (s, a) \in \mathcal{S} \times \mathcal{A} \tag{10}$$

$$\mu_{k,sa}^{(t)} = \frac{\sum_{\ell=1}^{k-1} \sum_{t \in E_{\ell,sa}} y_{k,\xi^-}^{(t+1)} + \beta \mu_{0,sa}}{N_{k,sa} + \beta}, \forall (s, a) \in \mathcal{S} \times \mathcal{A} \tag{11}$$

where $y_{k,\xi^-}^{(t+1)} = R_{k,t+1} + \max_{a' \in \mathcal{A}} f_{\theta_k^{(t+1)}}(S_{k,t+1}, a', \xi^-)$.

| Hyper-parameters | Finite MDP with Horizon $H$ |
|---|---|
| target network freq | 1 |
| sample update ratio | 1 |
| training freq | H |

Table 3: Hyper-parameters of our Tabular-HyperFQI

## B  ENVIRONMENT SETTINGS

In this section, we describe our environment used in experiments. We firstly use the DeepSea to demonstrate the exploration efficiency of our HyperFQI. DeepSea is a reward-sparse environment that demands extensive exploration (see Figure 5). The environment under consideration has a discrete action space consisting of two actions: moving left or right. During each run of the experiment, the action for moving right is randomly sampled from Bernoulli distribution for each row. Specifically, the action variable takes binary values of 1 or 0 for moving right, and the action map is different for each run of the experiment. The agent receives a reward of 0 for moving left, and a penalty of $-(0.01/N)$ for moving right, where $N$ denotes the size of DeepSea. The agent will earn a reward of 1 upon reaching the lower-right corner. The optimal policy for the agent is to learn to move continuously to the right. The sparse rewards and states present in this environment effectively showcase the exploration efficiency of our method without any additional complexity.

For the experiments on the Atari games, we evaluate our HyperFQI on 26 of the 55 games from the full ALE suite. We utilized the standard wrapper provided by OpenAI gym. For example, we terminated each environment after a maximum of 108K steps without using sticky actions. For further details on the settings used for the Atari games, please refer to the Table 4.

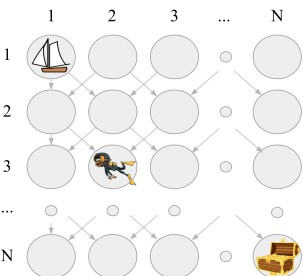

Figure 5: Illustration for DeepSea.

| Hyper-parameters | Setting |
|---|---|
| Grey-scaling | True |
| Observation down-sampling | (84, 84) |
| Frames stacked | 4 |
| Action repetitions | 4 |
| Reward clipping | [-1, 1] |
| Terminal on loss of life | True |
| Max frames per episode | 108K |

Table 4: Detailed settings for Atari games

## C  ADDITIONAL RESULTS

This section presents additional results for our HyperFQI algorithm. We demonstrate its robustness through ablation experiments on DeepSea. Moreover, we provide detailed results for each environment of Atari, highlighting the superiority of our approach over other baselines.

### C.1  RESULTS ON DEEPSEA

In Section 4.1, we noted that a larger $M$ in the Gaussian distributional index can harm the algorithm's performance due to increased difficulty in estimating the expectation. To address this, we can increase the number of indices $|\tilde{\Xi}|$ of Equation (5) during the update stage for each state in the batch, thereby improving the estimation of expectation. As shown in Figure 6 for $M = 16$, increasing $|\tilde{\Xi}|$ can lead to better performance. However, some seeds with $|\tilde{\Xi}| = 80$ still do not work well for $M = 16$. To achieve accurate estimation of expectation, $|\tilde{\Xi}|$ should grow exponentially with $M$, but this comes at the cost of increased computation, slowing down the algorithm. To balance performance and computation, we have chosen $M = 4$ and $|\tilde{\Xi}| = 20$ as our default hyper-parameters, which have demonstrated superior performance in Figure 2.

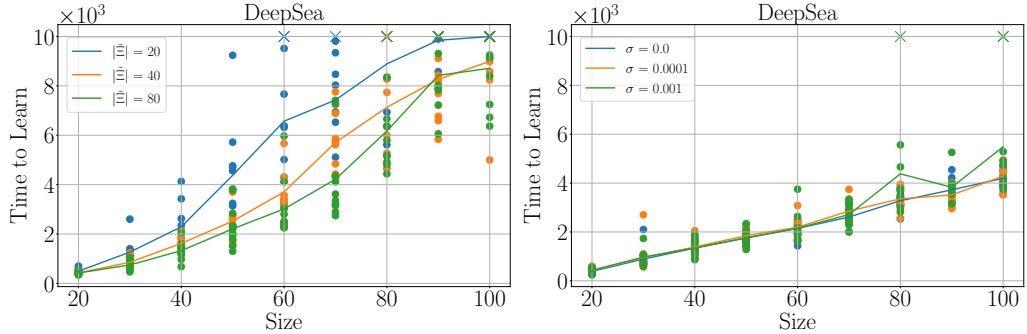

Figure 6: Ablation results under different $|\tilde{\Xi}|$.  Figure 7: Ablation results under different $\sigma$.

In addition, we have also investigated the effect of the $\sigma$ of Equation (3) on our methods, as shown in Figure 7. Our HyperFQI is not sensitive to this hyper-parameter, and we have selected $\sigma = 0.0001$ as our default hyper-parameters.

We observed that HyperFQI-OIS method performs better when using Gaussian distributional index, but not with one-hot index in Figure 2(b). This is because $M = 4$ cannot accurately estimate the expectation when using one-hot index. However, We observed that HyperFQI with one-hot index achieves good performance when $M = 16$ in Figure 3(b). Thus, we evaluated the efficiency of HyperFQI-OIS with one-hot index when $M = 16$. As shown in Figure 8, with an appropriate $M$ for one-hot index, our OIS actually achieves better performance. Through these experiments, we demonstrated the superiority of HyperFQI-OIS in both Gaussian distributional and one-hot index.

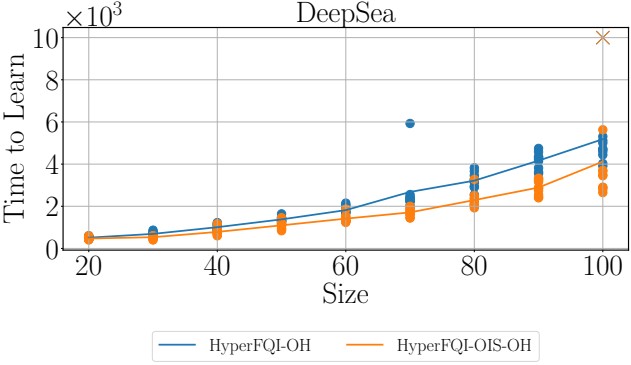

Figure 8: Results on DeepSea with one-hot index. In this experiment, we set $M = 16$.

To further illustrate the data efficiency of HyperFQI, we conducted a comparison with additional baselines on DeepSea. Specifically, we included BootDQN with a randomized prior function (Osband et al., 2018) as a baseline, and reproduced the results using https://github.com/google-deepmind/bsuite referring to it as BootDQN w. prior. We also applied optimistic index sampling (OIS) to this baseline, referred to as BootDQN-OIS w. prior. BootDQN-OIS can be regarded as an implementation of LSVI-PHE (Ishfaq et al., 2021) with DNN.[3] For both bootstrapped methods, we firstly set the number of ensembles to 4, consistent with the dimension of index used in HyperFQI. As depicted in the Figure 9(a), BootDQN w. prior fails to solve DeepSea with size 20 within $10e3$ episodes. Although OIS enhances exploration, BootDQN-OIS w. prior knowledge solves DeepSea with size 20 with in few seeds but fails in DeepSea with size 30. Furthermore, we compared our HyperFQI method to bootstrapped methods using 16 ensemble networks, as illustrated in Figure 9(b). While increasing the number of ensembles improves exploration, it remains insufficient for solving DeepSea with large size. In contrast, our HyperFQI approach efficiently solves DeepSea with just 4 dimensions of index, demonstrating its superiority.

---

[3]As we do not find the official implementation of LSVI-PHE (Ishfaq et al., 2021) anywhere. We use results of BootDQN-OIS to represent the performance of LSVI-PHE.

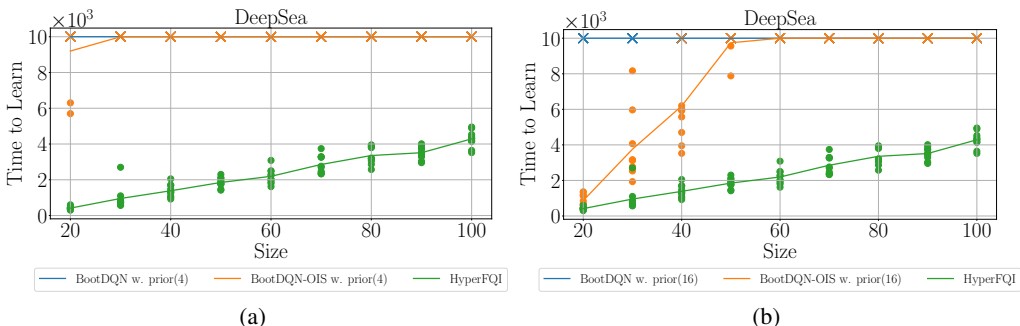

Figure 9: Compare results on DeepSea with more baselines.

## C.2 RESULTS ON ATARI

We demonstrated the efficiency of our HyperFQI in handling data and computation in Section 4.2. Here, we present comprehensive results on each environment to further establish the superiority of our approach.

In Table 5, we present the best score achieved in each environment with 2M steps. Our experimental protocol includes evaluating the best model 200 times for each seed, after completing training for each Atari game. We then calculate the average score from these 200 evaluations as the score for each seed. Finally, we calculate and report the average score across 20 seeds as the final score for each Atari game. The scores for Rainbow and DDQN are obtained from Hessel et al. (2018), which were based on 200M Frames. Specifically, we extracted the first 20M steps from these results to compare them with our HyperFQI. For HyperDQN, we refer to the results from (Li et al., 2022a) and similarly extracted the first 20M steps for comparison purposes. The DER was executed using the popular implementation available at `https://github.com/Kaixhin/Rainbow`. We conducted all experiments with 20 different seeds and computed the average best score with the best policy during training.

We also present the relative improvement of our HyperFQI in comparison to other baselines for each game, which is determined by the given following equation as per (Wang et al., 2016).

$$\text{relative improvement} = \frac{\text{proposed} - \text{baseline}}{\max(\text{human}, \text{baseline}) - \text{human}}$$

Our classification of environments into three groups, namely "hard exploration (dense reward)", "hard exploration (sparse reward)" and "easy exploration", is based on the taxonomy proposed by Bellemare et al. (2016). The overall results are illustrated in Figure 10, Figure 11, Figure 12 and Figure 13.

Our HyperFQI algorithm exhibits significant improvement compared to DoubleDQN, DER, and HyperDQN in environments with "easy exploration", and overall it performs better in all environments. This indicates that HyperFQI has better generalization and exploration abilities. On the other hand, when compared to Rainbow, our algorithm performs better in environments which are in the group of "hard exploration (dense reward)", demonstrating our superior deep exploration capabilities. However, in the case of Freeway, which belongs to the "hard exploration (sparse reward)" group, both HyperFQI and Rainbow achieve similar optimal scores (as shown in Table 5), suggesting no significant improvement in this environment. Overall, our HyperFQI showcases better generalization and exploration efficiency.

Figure 14 illustrates the learning curve for each game. Our HyperFQI has shown superior performance in comparison to DDQN (ours), attributed to the incorporation of a hypermodel that enhances exploration in Atari games. Additionally, our HyperFQI variants demonstrated stable and efficient learning, as indicated by the results. The learning curves of these variants exhibit remarkable similarity, indicating the robustness of our HyperFQI on Atari games. However, our experiments have demonstrated that the HyperFQI-OIS outperforms the others in DeepSea, which necessitates deep exploration. Furthermore, it is worth highlighting that the learning curve of our algorithm continues

| Game | Random | Human | DDQN | DER | Rainbow | HyperDQN | HyperFQI |
|---|---|---|---|---|---|---|---|
| Alien | 227.8 | 7127.7 | 722.7 | 1642.2 | 1167.1 | 862.0 | **1830.2** |
| Amidar | 5.8 | 1719.5 | 61.4 | 476.0 | 374.0 | 140.0 | **800.4** |
| Assault | 222.4 | 742.0 | 815.3 | 488.3 | 2725.2 | 494.2 | **3276.2** |
| Asterix | 210.0 | 8503.3 | 2471.1 | 1305.3 | 3213.3 | 713.3 | 2370.2 |
| BankHeist | 14.2 | 753.1 | 7.4 | 460.5 | 411.1 | 272.7 | 430.3 |
| BattleZone | 2360.0 | 37187.5 | 3925.0 | 19202.5 | 19379.7 | 11266.7 | **29399.0** |
| Boxing | 0.1 | 12.1 | 26.7 | 1.7 | 69.9 | 6.8 | **74.0** |
| Breakout | 1.7 | 30.5 | 2.0 | 6.5 | 137.3 | 11.9 | 54.8 |
| ChopperCommand | 811.0 | 7387.8 | 354.6 | 1488.9 | 1769.4 | 846.7 | **2957.2** |
| CrazyClimber | 10780.5 | 35829. | 53166.5 | 36311.1 | 110215.8 | 42586.7 | **121855.8** |
| DemonAttack | 152.1 | 1971.0 | 1030.8 | 955.3 | 45961.3 | 2197.7 | **5852.0** |
| Freeway | 0.0 | 29.6 | 5.1 | 32.8 | 32.4 | 30.9 | 32.2 |
| Frostbite | 65.2 | 4334.7 | 358.3 | 3628.3 | 3648.7 | 724.7 | **4583.9** |
| Gopher | 257.6 | 2412.5 | 569.8 | 742.1 | 4938.0 | 1880.0 | **7365.8** |
| Hero | 1027.0 | 30826.4 | 2772.9 | 15409.4 | 11202.3 | 9140.3 | 12324.7 |
| Jamesbond | 29.0 | 302.8 | 15.0 | 462.1 | 773.1 | 386.7 | **951.6** |
| Kangaroo | 52.0 | 035.0 | 134.9 | 8852.3 | 6456.1 | 3393.3 | 8517.1 |
| Krull | 1598.0 | 2665.5 | 6583.3 | 3786.7 | 8328.5 | 5488.7 | 8222.6 |
| KungFuMaster | 258.5 | 22736.3 | 12497.2 | 15457.0 | 25257.8 | 12940.0 | 23821.2 |
| MsPacman | 307.3 | 6951.6 | 1912.3 | 2333.7 | 1861.1 | 1305.3 | **3182.3** |
| Pong | -20.7 | 14.6 | -15.4 | 20.6 | 5.1 | 20.5 | 20.5 |
| PrivateEye | 24.9 | 69571.3 | 37.8 | 900.9 | 100.0 | 64.5 | 171.9 |
| Qbert | 163.9 | 13455.0 | 1319.4 | 12345.5 | 7885.3 | 5793.3 | 12021.9 |
| RoadRunner | 11.5 | 7845.0 | 3693.5 | 14663.0 | 33851.0 | 7000.0 | 28789.4 |
| Seaquest | 68.4 | 42054.7 | 367.6 | 662.0 | 1524.7 | 370.7 | **2732.4** |
| UpNDown | 533.4 | 11693.2 | 3422.8 | 6806.3 | 39187.1 | 4080.7 | 19719.2 |

Table 5: The best score over 200 evaluation episodes for the best policy in hindsight (after 2M steps) for Atari games. The performance of the random policy and the human expert is from dqn_zoo (Quan & Ostrovski, 2020).

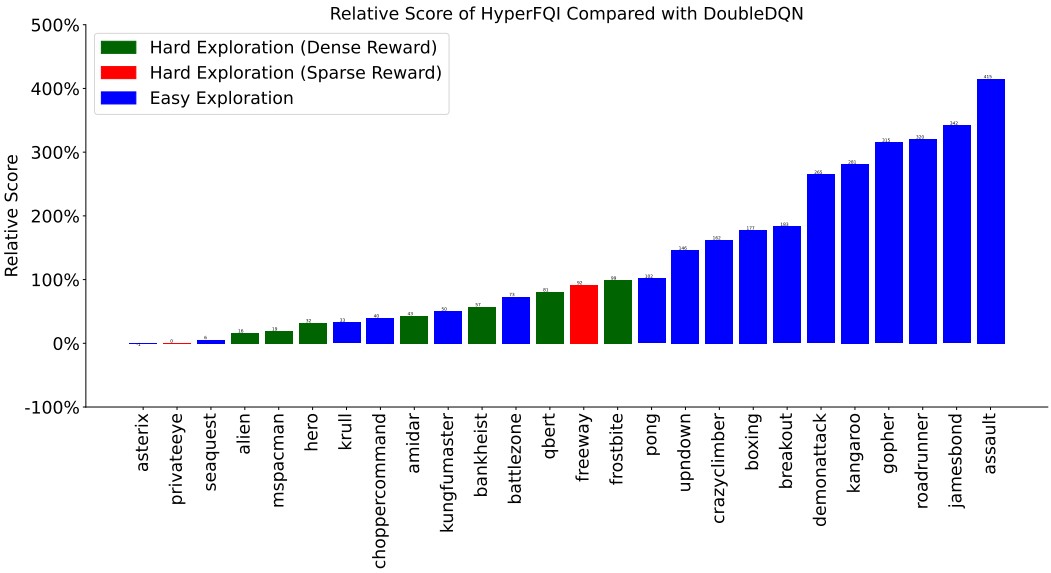

Figure 10: Relative improvement of HyperFQI compared with DoubleDQN

to rise in certain environments, indicating that our HyperFQI can achieve even better performance with additional training.

In addition, we demonstrated the superiority of our approach on the 8 hardest exploration Atari games with more baselines. We utilized the released results of AdamLMCDQN (Ishfaq et al., 2023), LangevinAdam (Ishfaq et al., 2023), and HyperDQN (Li et al., 2022a) on Atari. Additionally, we

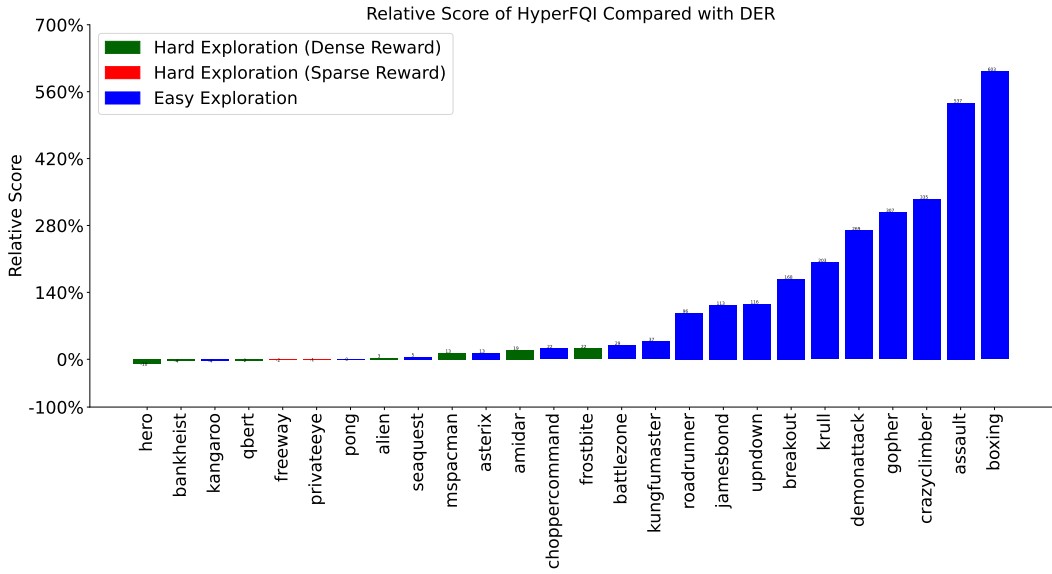

Figure 11: Relative improvement of HyperFQI compared with DER

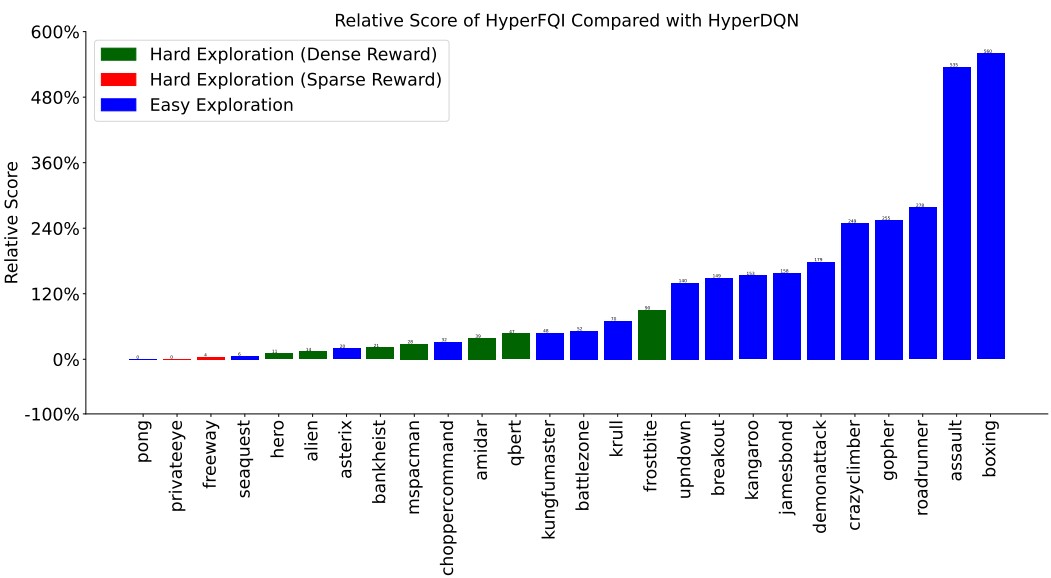

Figure 12: Relative improvement of HyperFQI compared with HyperDQN

adopted SANE (Aravindan & Lee, 2021) as our new baseline, which leverages a variational distribution to approximate the posterior. To reproduce the outcomes, we employed the official implementation of SANE. Given the time and resource constraints, we trained both HyperFQI and SANE with 5 different seeds, up to a limit of 2M steps. The Figure 15 indicates that our HyperFQI outperformed other baselines on 5 out of 8 games. For Solaris and Venture, we anticipate that providing more training time can further improve the performance of our HyperFQI. Overall, these results demonstrate the effectiveness and exploration ability of our HyperFQI in complex observation environments.

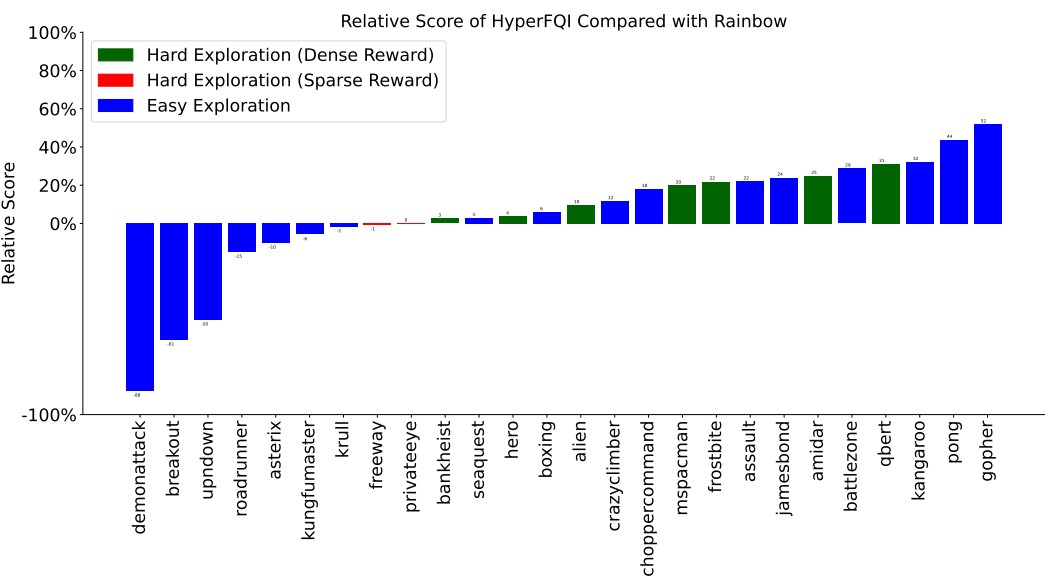

Figure 13: Relative improvement of HyperFQI compared with Rainbow

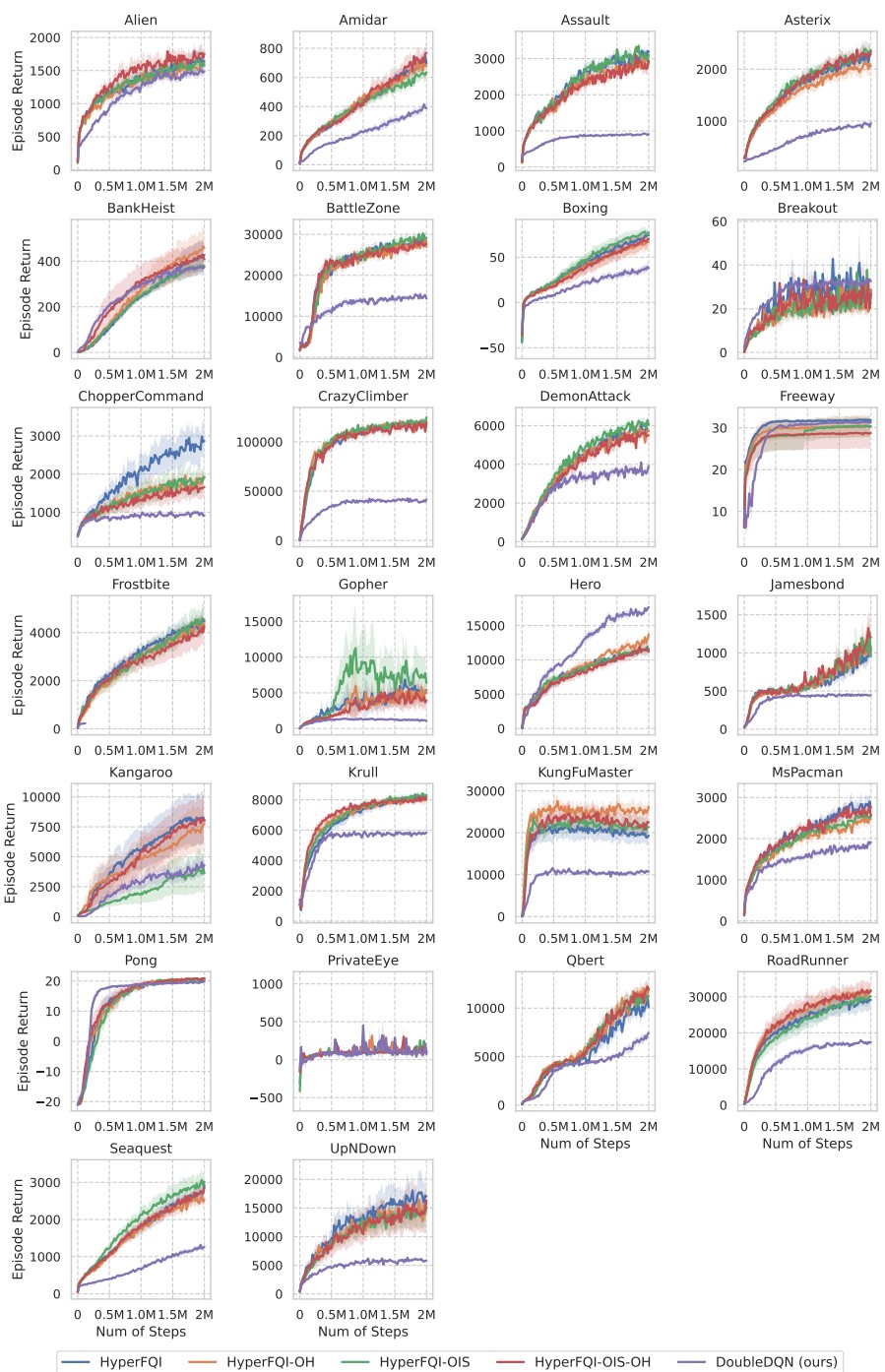

Figure 14: Learning curve for each game. All variants exhibit remarkable similarity, indicating the robustness of our HyperFQI on Atari games.

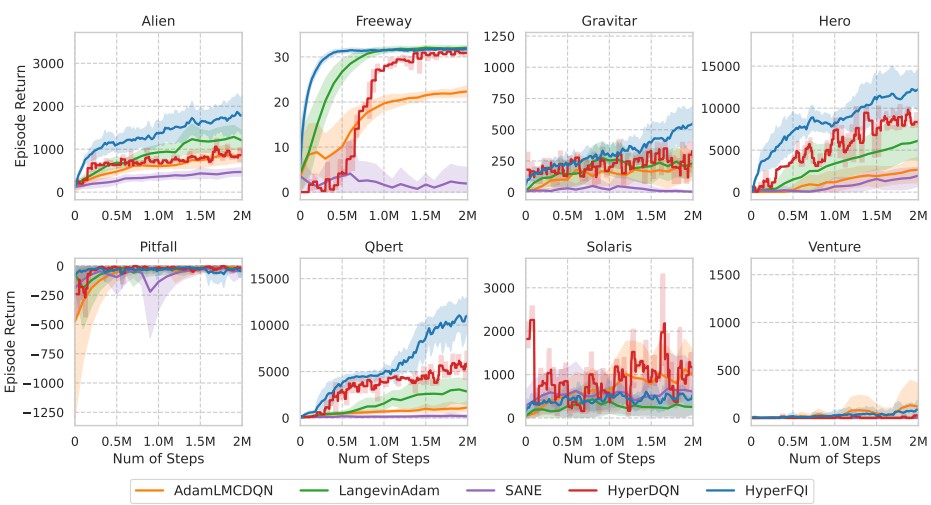

Figure 15: Comparative results on 8 hardest exploration games.

## D    PROBABILISTIC FORMALISM

One of the difficulties in the analysis is to deal with the sequential dependence structure among the random variables generated from the reinforcement learning problems. We define some important concept that would be useful in the analysis.

Let $(\Omega, \mathcal{F}, (\mathcal{F}_t)_{t \geq 0}, \mathbb{P})$ be a complete filtered probability space.

**Definition D.1** (Adapted process)**.** For an index set $I$ of the form $\{t \in \mathbb{N} : t \geq t_0\}$ for some $t_0 \in \mathbb{N}$, we say a stochastic process $(\mathrm{x}_t)_{t \in I}$ is adapted to the filtration $(\mathcal{F}_t)_{t \in I}$ if each $\mathrm{x}_t$ is $\mathcal{F}_t$-measurable.

**Definition D.2** ((Conditionally) $\sigma$-sub-Gaussian)**.** We first describe the property associated with one-dimensional random variable. Second, we describe the generalization in high-dimension random vector.

- Random variables
  - We say a random variable x is $\sigma$-sub-Gaussian if
  $$\mathbb{E}[\exp(\lambda \mathrm{x})] \leq \exp\left(\frac{\lambda^2 \sigma^2}{2}\right), \qquad \forall \lambda \in \mathbb{R}.$$

  - Let $(\mathbf{x}_t)_{t \geq 1} \subset \mathbb{R}^M$ be a stochastic process adapted to filtration $(\mathcal{F}_t)_{t \geq 1}$. Let $\sigma = (\sigma_t)_{t \geq 0}$ be a stochastic process adapted to filtration $(\mathcal{F}_t)_{t \geq 0}$. We say the process is $(\mathbf{x}_t)_{t \geq 1}$ is conditionally $\sigma$-sub-Gaussian if
  $$\mathbb{E}[\exp(\lambda \mathbf{x}_t) \mid \mathcal{F}_{t-1}] \leq \exp\left(\frac{\lambda^2 \sigma_{t-1}^2}{2}\right), \quad a.s. \quad \forall \lambda \in \mathbb{R}.$$

  Specifically for the index $t+1$, we can say $\mathbf{x}_{t+1}$ is ($\mathcal{F}_t$-conditionally) $\sigma_t$-sub-Gaussian. If $\sigma_t$ is a constant $\sigma$ for all $t \geq 0$, then we just say (conditionally) $\sigma$-sub-Gaussian.

- Random vectors
  - For random vector $\mathbf{x}$ or vector process $(\mathbf{x}_t)_{t \geq 1}$, we say it is $\sigma$-sub-Gaussian is for every fixed $v \in \mathbb{S}^{M-1}$ if the random variable $\langle v, \mathbf{x} \rangle$ or stochastic process $\langle v, \mathbf{x}_t \rangle$ is $\sigma$-sub-Gaussian.

**Definition D.3** (Almost sure unit-norm)**.** We say a random vector $\mathbf{x}$ is almost sure unit-norm if $\|\mathbf{x}\|_2 = 1$ almost surely.

**Definition D.4** ($c_x$-bounded process)**.** For an index set $I$ of the form $\{t \in \mathbb{N} : t \geq t_0\}$ for some $t_0 \in \mathbb{N}$, the stochastic process $(\mathrm{x}_t)_{t \in I}$ is $c_x$-bounded if $\mathrm{x}_t^2 \leq c_x$ almost surely for all $t \in I$.

## E    REGRET BOUND

Denote the regret of a policy $\pi_k$ over episode $k$ by
$$\Delta_k := \mathbb{E}_{M,\mathrm{alg}}[V_M^{\pi^*}(s_{k,0}) - V_M^{\pi_k}(s_{k,0}],$$
where $\pi^*$ is an optimal policy for $M$. The goal of the agent is equivalent to minimizing the expected total regret up to episode $K$
$$\mathrm{Regret}(K, \mathrm{alg}) := \mathbb{E}_{\mathrm{alg}} \sum_{k=1}^{K} \Delta_k, \tag{12}$$

where the subscript $\mathrm{alg}$ under the expectation indicates that policies are generated through algorithm $\mathrm{alg}$. Note that the expectation in Equation (12) is over the random transitions and rewards, the possible randomization in the learning algorithm $\mathrm{alg}$, and also the unknown MDP $M$ based on the agent designer's prior beliefs.

**Assumption E.1** (Finite-horizon time-inhomogeneous MDPs)**.** We consider a problem class that can be formulated as a special case of the general formulation in Section 2. Assume the state space factorizes as $\mathcal{S} = \mathcal{S}_0 \cup \mathcal{S}_1 \cup \mathcal{S}_2 \cup \cdots \cup \mathcal{S}_{H-1}$ where the state always advances from some state $s_t \in \mathcal{S}_t$ to $s_{t+1} \in \mathcal{S}_{t+1}$ and the process terminates with probability 1 in period $H$. For notational convenience, we assume each set $\mathcal{S}_0 = \ldots = \mathcal{S}_{H-1} = \mathcal{X}$ contains an equal number of elements that is $\mathcal{X}$. That is $|\mathcal{S}| = |\mathcal{X}|H$.

We study the regret of HyperFQI under the following Bayesian model for the MDP $M$.

**Assumption E.2** (Independent Dirichlet prior for outcomes). For each $(s, a) \in \mathcal{S} \times \mathcal{A}$, the outcome distribution is drawn from a Dirichlet prior

$$P_{sa} \sim \text{Dirichlet}(\alpha_{0,sa})$$

for $\alpha_{0,sa} \in \mathbb{R}_+^{\mathcal{S}}$ and each $P_{sa}$ is drawn independently across $(s, a)$. Assume there is $\beta \geq 3$ such that $\mathbf{1}^\top \alpha_{0,sa} = \beta$ for all $(s, a)$.

A key observation that enable the regret analysis is that hypermodel can approximate the posterior distribution of the optimal $Q^*$-values with low computation cost. This is formalized in the following.

**Lemma E.3** (Approximate posterior variance (restated)). *For $m_k$ defined in Equation (6) with $\mathbf{z} \sim$ Uniform$(\mathbb{S}^{M-1})$. For any $k \geq 1$, a good event $\mathcal{G}_k(s, a)$ is defined as*

$$\mathcal{G}_k(s, a) = \left\{ \|m_k(s, a)\|^2 \in \left( \frac{\sigma^2}{N_{k,sa} + \beta}, \frac{3\sigma^2}{N_{k,sa} + \beta} \right) \right\}.$$

*Then the joint event $\cap_{(s,a,k) \in \mathcal{S} \times \mathcal{A} \times [K]} \mathcal{G}_k(s, a)$ holds w.p. at least $1 - \delta$ if $M \simeq \log(SAK/\delta)$.*

To prove Lemma 5.2, we develop a new probability tools for sequential random projection in Appendices F and G. This is a technical contribution, which maybe of independent interest. Once the approximation lemma is established, the rest proof can be reduced to the Bayesian analysis of RLSVI in Osband et al. (2019b).

We want to emphsize a key argument that enables efficient deep exploration is the stochastic optimism of HyperFQI.

**Definition E.4** (Stochastic optimism). A random variable $X$ is stochastically optimistic with respect to another random variable $Y$, written $X \geq_{SO} Y$, if for all convex increasing functions $u : \mathbb{R} \to \mathbb{R}$

$$\mathbb{E}[u(X)] \geq \mathbb{E}[u(Y)].$$

We show that HyperFQI is stochastic optimistic in the sense that it overestimates the value of each action in expectation. This is formalized in the following lemma.

**Proposition E.5.** *If Assumption E.2 holds and HyperFQI is applied with parameters parameters $(M, \mu_0, \sigma, \sigma_0)$ satisfying $M \simeq \log(SAK)$, $(\sigma^2/\sigma_0^2) = \beta$, $\sigma \geq \sqrt{3}H$ and $\min_{s,a} \mu_{0,s,a} \geq H$,*

$$f_{\theta_k}(s, a, \xi) \,|\, \mathcal{H}_k \geq_{SO} Q_M^*(s, a) \,|\, \mathcal{H}_k. \tag{13}$$

*for any history $\mathcal{H}_k$ and state-action pair $(s, a) \in \mathcal{S}_0 \times \mathcal{A}$ given the event $\mathcal{A}_k$ defined in Lemma 5.2 holds.*

# F    PROOF OF THE KEY APPROXIMATE POSTERIOR LEMMA 5.2

Now we provide the key theorem that enable the whole analysis. This is a novel probability tool for sequential random projection. We use short notation for $[n] = \{1, 2, \ldots, n\}$ and $\mathcal{T} = \{0, 1, \ldots, T\} = \{0\} \cup [T]$.

**Theorem F.1** (Sequential random projection in adaptive process). *Let $(\mathcal{F}_t)_{t \geq 0}$ be a filtration. For any fixed $\varepsilon \in (0, 1)$ any fixed $s \in \mathbb{R}_+$, let $\mathbf{s} \in \mathbb{R}^M$ be an $\mathcal{F}_0$-measurable random vector satisfies $\mathbb{E}[\|\mathbf{s}\|^2] = s^2$ and $|\|\mathbf{s}\|^2 - s^2| \leq (\varepsilon/2)s^2$.*

*Let $(\mathbf{z}_t)_{t \geq 1} \subset \mathbb{R}^M$ be a stochastic process adapted to filtration $(\mathcal{F}_t)_{t \geq 1}$ such that it is $\sqrt{c_0/M}$-sub-Gaussian and each $\mathbf{z}_t$ is unit-norm. Let $(x_t)_{t \geq 1} \subset \mathbb{R}$ be a stochastic process adapted to filtration $(\mathcal{F}_{t-1})_{t \geq 1}$ such that it is $c_x$-bounded. Here, $c_0$ and $c_x$ are absolute constants.*

*If the following condition is satisfied*

$$M \geq \frac{16c_0(1 + \varepsilon)}{\varepsilon^2} \left( \log\left(\frac{1}{\delta}\right) + \log\left(1 + \frac{c_x T}{s^2}\right) \right),$$

*we have, with probability at least $1 - \delta$*

$$\forall t \in \mathcal{T}, \quad (1 - \varepsilon)\left( s^2 + \sum_{i=1}^t x_i^2 \right) \leq \|\mathbf{s} + \sum_{i=1}^t x_i \mathbf{z}_i\|^2 \leq (1 + \varepsilon)\left( s^2 + \sum_{i=1}^t x_i^2 \right).$$

*Remark* F.2. We say this is an "sequential random projection" argument because one can relate Theorem F.1 to the traditional random projection setting where $\Pi = (\mathbf{z}_1, \ldots, \mathbf{z}_T) \in \mathbb{R}^{M \times T}$ is a random projection matrix and $\boldsymbol{x} = (x_1, \ldots, x_T)^\top \in \mathbb{R}^T$ is the vector to be projected. When $s = 0$ and $\mathbf{s} = 0$, this is essentially an analog of distributional JL lemma (discribed in Lemma H.3) while the traditional JL lemma are NOT handle the sequential dependence structure in our setup. Therefore, Theorem F.1 also an innovation in the literature of random projection and sequential analysis.

*Remark* F.3. The unit-norm condition in the Theorem F.1 is easy to remove. Then, more distribution of random vectors can be covered in our probability framework. We leave it for the future work.

**Example F.4** (Stylized stochastic process satisfying the condition in Theorem F.1.)**.** If $\mathbf{s}$ is a random vector that is independent with all following random variables and $(\mathbf{z}_t)_{t \geq 1}$ are i.i.d random vectors, each sampled from $\mathcal{U}(\mathbb{S}^{M-1})$. The stochastic process $(x_t)_{t \geq 1}$ has the following dependence structure with the process $(\mathbf{z}_t)_{t \geq 1}$:

- $x_t$ is dependent on $\mathbf{s}, x_1, \mathbf{z}_1, \ldots, x_{t-1}, \mathbf{z}_{t-1}$.

- $\mathbf{z}_t$ is independent of $\mathbf{s}, x_1, \mathbf{z}_1, \ldots, x_{t-1}, \mathbf{z}_{t-1}, x_t$

Define the filtration $(\mathcal{F}_t)_{t \geq 0}$ where $\mathcal{F}_t = \sigma(\mathbf{s}, x_1, \mathbf{z}_1, \ldots, x_t, \mathbf{z}_t, x_{t+1})$. From Example I.2, we notice $\mathbf{z} \sim \mathcal{U}(\mathbb{S}^{M-1})$ is $(1/\sqrt{M})$-sub-Gaussian random vector. Thus, $(\mathbf{z}_t)_{t \geq 1} \overset{i.i.d.}{\sim} \mathcal{U}(\mathbb{S}^{M-1})$ is a stochastic process adapted to $(\mathcal{F}_t)_{t \geq 1}$ and is $1/\sqrt{M}$-sub-Gaussian and unit-norm.

*Proof of Lemma 5.2.* Take a look at the formula in Equation (6) and apply Theorem F.1 with sequence $(\mathbf{z}_{\ell,t})_{\ell \geq 1, \geq H-1 t \geq 0}$ and $(x_{\ell,t})_{\ell \geq 1, \geq H-1 t \geq 0}$ s.t. $x_{\ell,t} = \sigma \mathbb{1}_{t \in E_{\ell,sa}}$ and $\mathbf{s} = \beta \sigma_0 \mathbf{z}_{0,sa}, s = \beta \sigma_0$ for each state action pair $(s, a) \in \mathcal{S} \times \mathcal{A}$. Then taking union bound over the set $\mathcal{S} \times \mathcal{A}$ yields the results. $\qquad \square$

## G  PROOF OF SEQUENTIAL RANDOM PROJECTION

In this section, we describe our technical innovation in a probability statement for sequential random projection. Based on a novel and careful construction of stopped process that controls the deviation behavior of the good event on concentration, we adopt the method of mixtures (Peña et al., 2009) in self-normalized process to derive a probability tool stated in Theorem F.1. This bound is new to the whole literature of random projection and also sequential analysis, which may be of independent interest.

First, we state a Peña et al. (2009) type self-normalized bound that would be useful to prove our main theoretical contribution of sequential random projection in Theorem F.1.

**Theorem G.1** (Any-time self-normalized concentration bound)**.** *Let* $(\mathcal{F}_t)_{t \geq 0}$ *be a filtration and* $\{(A_t, B_t), t \geq 1\}$ *be a sequence of pairs of random variables satisfying that for all* $\lambda \in \mathbb{R}$

$$\left\{ \exp\left( \lambda A_t - \frac{\lambda^2}{2} B_t^2 \right), \mathcal{F}_t, t \geq 1 \right\} \text{ is a supermartingale with mean} \leq 1. \tag{14}$$

*Then, for any fixed positive sequence* $(L_t)_{t \geq 1}$*, with probability at least* $1 - \delta$

$$\forall t \geq 1, \quad |A_t| \leq \sqrt{2 \left( B_t^2 + L_t \right) \log \left( \frac{1}{\delta} \frac{\left( B_t^2 + L_t \right)^{1/2}}{L_t^{1/2}} \right)} \tag{15}$$

The proof of Theorem G.1 can be found in Appendix G.2.

### G.1  SEQUENTIAL RANDOM PROJECTION ARGUMENT IN THEOREM F.1

Before dig into the proof, we identify some important sequence structure and also clarity our proof idea in a intuitive level.

### G.1.1    PREPARATION FOR THE PROOF OF THEOREM F.1

For $t \in \mathcal{T}$, let the short notation be

$$Y_t = \|\mathbf{s} + \sum_{i=1}^{t} x_i \mathbf{z}_i\|^2 - \left(s^2 + \sum_{i=1}^{t} x_i^2\right).$$

and $S_t = s^2 + \sum_{i=1}^{t} x_i^2$. Our *key observation* is that for any $t \in [T]$

$$\|\mathbf{s} + \sum_{i=1}^{t} x_i \mathbf{z}_i\|^2 = \|\mathbf{s} + \sum_{i=1}^{t-1} x_i \mathbf{z}_i + x_t \mathbf{z}_t\|^2$$

$$= \|\mathbf{s} + \sum_{i=1}^{t-1} x_i \mathbf{z}_i\|^2 + 2 \left(\mathbf{s} + \sum_{i=1}^{t-1} x_i \mathbf{z}_i\right)^{\top} x_t \mathbf{z}_t + x_t^2 \|\mathbf{z}_t\|^2$$

and thus we have the following relationship between $Y_t$ and $Y_{t-1}$

$$Y_t - Y_{t-1} = 2 x_t \mathbf{z}_t^{\top} (\mathbf{s} + \sum_{i=1}^{t-1} x_i \mathbf{z}_i) + x_t^2 \left(\|\mathbf{z}_t\|^2 - 1\right).$$

Since $\mathbf{z}_t$ is unit-norm, we can further simplify the exposition

$$Y_t - Y_{t-1} = 2 x_t \mathbf{z}_t^{\top} (\mathbf{s} + \sum_{i=1}^{t-1} x_i \mathbf{z}_i). \tag{16}$$

Another key observation is that the difference term in Equation (16) depends on the $(\mathbf{s} + \sum_{i=1}^{t-1} x_i \mathbf{z}_i)$ that is $F_{t-1}$-measurable. We can control the deviation of the difference $Y_t - Y_{t-1}$ by if we already have information in the history-dependent term. Intuitively, once the concentration behavior is bad, it is highly possible to be bad for the later time index. To mathematically formalize this intuition, we introduce a definition of good event and stopping time for analysis.

**Definition G.2** (Good event). For each time $t \in \mathcal{T}$, we introduce the good event $E_t$ under which the strongly concentration behavior is guaranteed, suppose $\varepsilon \in (0, 1)$,

$$E_t(\varepsilon) = \left\{ (1 - \varepsilon) \left(s^2 + \sum_{i=1}^{t} x_i^2\right) \le \|\mathbf{s} + \sum_{i=1}^{t} x_i \mathbf{z}_i\|^2 \le (1 + \varepsilon) \left(s^2 + \sum_{i=1}^{t} x_i^2\right) \right\}.$$

With short notation,

$$E_t(\varepsilon) = \{|Y_t| \le \varepsilon S_t\}.$$

We also define the stopping time as the first time the bad event happens, i.e. the good event in Definition G.2 violates.

**Definition G.3** (Stopping time). For any fixed $\varepsilon$, we define the stopping time

$$\tau(\varepsilon) = \min\{t \in \mathcal{T} : \neg E_t(\varepsilon)\}.$$

Based on the stopping time, we construct a **stopped** process. For $t \in [T]$, define the stopped difference term

$$X_t^{\tau} = (Y_t - Y_{t-1}) \mathbb{1}_{t \le \tau} \tag{17}$$

such that the process $(X_t^{\tau})_{t \ge 1}$ is adapted to the filtration $(\mathcal{F}_t)_{t \ge 1}$.

**Claim G.4.** *The stopping time $\tau$ defined in Definition G.3. Let $(X_t^{\tau})_{t \ge 1}$ be the stochastic process defined in Equation (17) which is adapted to $(\mathcal{F}_t)_{t \ge 0}$. Let $A_t^{\tau} = \sum_{i=1}^{t} X_i^{\tau}$. Further denote $(B_t^{\tau})^2 = \sum_{i=1}^{t} (C_i^{\tau})^2$ with*

$$(C_t^{\tau})^2 := \frac{4 c_0}{M} x_t^2 (1 + \varepsilon) S_{t-1} \mathbb{1}_{t \le \tau}.$$

*If the $(\mathcal{F}_t)_{t \ge 1}$-adapted process $(\mathbf{z}_t)_{t \ge 1}$ is $(\sqrt{\frac{c_0}{M}})$-sub-Gaussian and each $\mathbf{z}_t$ is unit-norm, then for any fixed $\lambda \in \mathbb{R}$*

$$M_t^{\tau}(\lambda) = \exp\left(\lambda A_t^{\tau} - \frac{\lambda^2}{2} (B_t^{\tau})^2\right), \quad t \ge 1$$

*is a supermartingale.*

*Proof of Claim G.4.* Note $\mathbb{1}_{t\leq\tau} = 1 - \mathbb{1}_{\tau\leq t-1}$ is $\mathcal{F}_{t-1}$-measurable. Thus, the vector $(\mathbf{s} + \sum_{i=1}^{t-1} x_i\mathbf{z}_i)\mathbb{1}_{t\leq\tau}x_t$ is $\mathcal{F}_{t-1}$-measurable. By the condition on process $(\mathbf{z}_t)_{t\geq 1}$, we conclude from the definition of conditionally sub-Gaussian from Definition D.2 that

$$
\begin{aligned}
\mathbb{E}[\exp(\lambda X_t^\tau) \mid \mathcal{F}_{t-1}] &= \mathbb{E}[\exp(2\lambda x_t\langle\mathbf{z}_t, \mathbf{s} + \sum_{i=1}^{t-1} x_i\mathbf{z}_i\rangle\mathbb{1}_{t\leq\tau}) \mid \mathcal{F}_{t-1}] \\
&\leq \exp\left(\frac{\lambda^2}{2}(4c_0/M)x_t^2\|\mathbf{s} + \sum_{i=1}^{t-1} x_i\mathbf{z}_i\|^2\mathbb{1}_{t\leq\tau}\right) \\
&\leq \exp\left(\frac{\lambda^2}{2}(4c_0/M)x_t^2(1+\varepsilon)S_{t-1}\mathbb{1}_{t\leq\tau}\right) \\
&= \exp\left(\frac{\lambda^2}{2}(C_t^\tau)^2\right)
\end{aligned}
$$

where the last inequality is because of the stopping time argument. $\qquad\square$

We also need the following lemma in the intial treatment of the proof of Theorem F.1.

**Lemma G.5** (Trigger lemma). *For any sequence of event $(\mathcal{E}_t, t \in \mathcal{T})$, define the stopping time $\tau$ as the first time $t$ the event $\mathcal{E}_t$ is violated, i.e.*

$$
\tau = \min\{t \in \mathcal{T} : \neg\mathcal{E}_t\}.
$$

*Then, the following equality holds for all $t \in \mathcal{T}$,*

$$
\{\tau \leq t\} = \neg\mathcal{E}_{t\wedge\tau}. \tag{18}
$$

### G.1.2 PROOF OF THEOREM F.1

Now we are ready to the proof.

*Proof of Theorem F.1.* We apply Lemma G.5 for $\mathcal{E}_t = E_t(\varepsilon)$ and it follows

$$
\begin{aligned}
\mathbb{P}\left(\exists t \in \mathcal{T}, \neg E_t(\varepsilon)\right) &= \mathbb{P}(\tau \leq T) = \mathbb{P}\left(\neg E_{T\wedge\tau}(\varepsilon)\right) \\
&= \mathbb{P}\left(|Y_{T\wedge\tau}| \geq \varepsilon S_{T\wedge\tau}\right) \\
&= \mathbb{P}\left(|Y_0 + \sum_{t=1}^{T}(Y_t - Y_{t-1})\mathbb{1}_{t\leq\tau}| \geq \varepsilon S_{T\wedge\tau}\right) \tag{19}
\end{aligned}
$$

By the construction of stopped process $Y_{T\wedge\tau} - Y_0 = \sum_{t=1}^{T} X_t^\tau = A_T^\tau$. Then, *our goal*, from Equation (19), becomes to upper bound the RHS of Equation (20),

$$
\mathbb{P}\left(\exists t \in \mathcal{T}, (\neg E_t)\right) = \mathbb{P}\left(|Y_0 + A_T^\tau| \geq \varepsilon S_{T\wedge\tau}\right) \tag{20}
$$

By Claim G.4, the process $(A_t^\tau, B_t^\tau)_{t\geq 1}$ with $A_t^\tau = \sum_{i=1}^{t} X_i^\tau = \sum_{i=1}^{t}(Y_t - Y_{t-1})\mathbb{1}_{t\leq\tau}$ and $(B_t^\tau)^2 = \sum_{i=1}^{t}(4c_0/M)x_t^2(1+\varepsilon)S_{t-1}\mathbb{1}_{t\leq\tau}$ satisfy the condition of Theorem G.1. Then we can apply the Theorem G.1: with probability at least $1 - \delta$,

$$
\forall t \geq 1, |A_t^\tau| \leq \sqrt{2\left((B_t^\tau)^2 + L_t\right)\log\left(\frac{1}{\delta}\frac{\left((B_t^\tau)^2 + L_t\right)^{1/2}}{L_t^{1/2}}\right)}
$$

Since by the condition in Theorem F.1, we have $|Y_0| \leq (\varepsilon/2)s^2$. Now we want to argue that for any fixed $\varepsilon \in (0, 1)$, with suitable choice of $L_T$ and $M$, we have with probability at least $1 - \delta$

$$
|Y_0 + A_T^\tau| \leq \underbrace{\sqrt{2\left((B_T^\tau)^2 + L_T\right)\log\left(\frac{1}{\delta}\frac{\left((B_T^\tau)^2 + L_T\right)^{1/2}}{L_T^{1/2}}\right)} + (\varepsilon/2)s^2}_{(I)} \leq \varepsilon S_{T\wedge\tau}. \tag{21}
$$

**Claim G.6.** *With some computations, we found the following configuration suffices for Equation* (21)*:*

$$L_T \leq \frac{4(1+\varepsilon)s^4}{2M} \quad and \quad M \geq (16(1+\varepsilon)/\varepsilon^2)\left(\log\left(\frac{1}{\delta}\right) + \log\left(1 + \frac{T}{s^2}\right)\right).$$

*Proof of Claim G.6.* Recall the definition

$$S_t = s^2 + \sum_{t=1}^{t} x_i^2$$

We first calculate the term $(B_T^\tau)^2$ by our construction:

$$(B_T^\tau)^2 \leq \frac{4c_0}{M}\sum_{t=1}^{T \wedge \tau} x_t^2\left((1+\varepsilon)S_{t-1}\right)$$

$$= \frac{4c_0(1+\varepsilon)}{M}\sum_{t=1}^{T \wedge \tau} x_t^2\left(S_{T \wedge \tau} - (S_{T \wedge \tau} - S_{t-1})\right)$$

$$\leq \frac{4c_0(1+\varepsilon)}{M}(S_{T \wedge \tau} - s^2)S_{T \wedge \tau}$$

Then, the almost sure upper bound of $(B_T^\tau)^2$ assuming $x_t^2 \leq c_x$ is

$$(B_T^\tau)^2 \leq \frac{4c_0(1+\varepsilon)}{M}\sum_{t=1}^{T}(s^2 + (t-1)c_x) \leq \frac{4c_0(1+\varepsilon)}{M}(s^2 T + c_x T^2/2)$$

Since $(a+b)^2 \leq (1+\lambda)(a^2 + (1/\lambda)b^2)$ for all $\lambda$,

$$(I)^2 \leq (1+\lambda)\left(2\left(B_T^2 + L_T\right)\log\left(\frac{1}{\delta}\frac{\left(B_T^2 + L_T\right)^{1/2}}{L_T^{1/2}}\right) + \frac{\varepsilon^2 s^4}{4\lambda}\right)$$

Let $L_T = c\ell/M$ and $c = 4c_0(1+\varepsilon)$ and $\ell$ to be determined.

$$(I)^2 \leq (1+\lambda)\left(2\left(B_T^2 + L_T\right)\log\left(\frac{1}{\delta}\frac{\left(B_T^2 + L_T\right)^{1/2}}{L_T^{1/2}}\right) + \frac{\varepsilon^2 s^4}{4\lambda}\right)$$

$$\leq (1+\lambda)\left(\frac{2c}{M}\left((S_{T \wedge \tau} - s^2)S_{T \wedge \tau} + \ell\right)\log\left(\frac{1}{\delta}\sqrt{\frac{(s^2 T + c_x T^2/2 + \ell)}{\ell}}\right) + \frac{\varepsilon^2 s^4}{4\lambda}\right)$$

Let $M = (2c/m)\log\left(\frac{1}{\delta}\sqrt{\frac{(s^2 T + c_x T^2/2) + \ell}{\ell}}\right)$, we can simplify

$$(I)^2 \leq (1+\lambda)\left(m((S_{T \wedge \tau} - s^2)S_{T \wedge \tau} + \ell) + \frac{\varepsilon^2 s^4}{4\lambda}\right)$$

Let $\ell = s^4/2c_x$, $m = \varepsilon^2/(1+\lambda)$ and $\lambda = 1$, we have

$$(I)^2 \leq \varepsilon^2((S_{T \wedge \tau} - s^2)S_{T \wedge \tau} + s^4/2 + s^4/2) \leq \varepsilon^2 S_{T \wedge \tau}^2$$

where the last inequality is because $s^2 = S_0 \leq S_{T \wedge \tau}$ and $s^4 \leq s^2 S_{T \wedge \tau}$. The conclusion is that we could select

$$M \geq (16c_0(1+\varepsilon)/\varepsilon^2)\log\left(\frac{1}{\delta}\sqrt{\frac{2s^2 c_x T + c_x^2 T^2 + s^4}{s^4}}\right)$$

$$= (16c_0(1+\varepsilon)/\varepsilon^2)\left(\log\left(\frac{1}{\delta}\right) + \log\left(1 + \frac{c_x T}{s^2}\right)\right)$$

and the auxiliary variable

$$L_T \leq \frac{4c_0(1+\varepsilon)s^4}{2Mc_x}.$$

□

□

## G.2 PROOF OF THEOREM G.1: METHOD OF MIXTURES

Robbins-Siegmund method of mixtures (Robbins & Siegmund, 1970) originally is developed to evaluate boundary crossing probabilities for Brownian motion. The method was further developed in the general theory for self-normalized process (de la Peña, Klass, and Lai, 2004; Peña, Lai, and Shao, 2009; Lai, 2009).

*Remark* G.7 (Essential idea of Laplace approximation). If we integrate the exponential of a function that has a pronounced maximum, then we can expect that the integral will be close to the exponential function of the maximum. In our case, let

$$M_t(\lambda) = \exp\left(\lambda A_t - \frac{\lambda^2}{2} B_t^2\right)$$

Informally, with this principle of Laplace approximation, we would have

$$\max_\lambda M_t(\lambda) \approx \int_\Omega M_t(\lambda) dh(\lambda)$$

where $h$ is some measure on $\Omega$.

*The main benefit of replacing the maximum $\max_\lambda M_t(\lambda)$ with an integral $\bar{M}_t := \int_\Omega M_t(\lambda) dh(\lambda)$ is that we can handle the expectation $\mathbb{E}[\bar{M}_t]$ easier while we don't know the upper bound on $\mathbb{E}[\max_\lambda M_t(\lambda)]$. This is formalized in the following lemma.*

**Lemma G.8.** *Let $(h_t)$ be a sequence of probability measures on $\Omega$. If $(M_t(\lambda), \mathcal{F}_t, t \geq 1)$ is a supermartingale with $\mathbb{E}[M_1(\lambda)] \leq 1$ for all $\lambda \in \Omega$, then for any $t \geq 1$, the integrated random variable $\bar{M}_t = \int_\Omega M_t(\lambda) dh_t(\lambda)$ has expectation $\mathbb{E}[\bar{M}_t] \leq 1$.*

*Further, let $\tau$ be a stopping time with respect to filtration $(\mathcal{F}_t)_{t\geq 0}$, i.e. $\{\tau \leq t\} \in \mathcal{F}_t, \forall t \geq 0$. Then $M_\tau(\lambda)$ is almost surely well-defined with expectation $\mathbb{E}[M_\tau(\lambda)] \leq 1$ as well as $\mathbb{E}[\bar{M}_\tau] \leq 1$.*

*Proof.* Using Fubini's theorem and the fact that $M_t(\lambda)$ is a supermartingale with $\mathbb{E}[M_t(\lambda)] \leq \mathbb{E}[M_1(\lambda)] = 1$, we have

$$\mathbb{E}[\bar{M}_t] = \int_\Omega \mathbb{E}[M_t(\lambda)] dh_t(\lambda) \leq 1.$$

For the expectation of stopped version $M_\tau(\lambda)$ and $\bar{M}_\tau$, we apply (supermartingale) optional sampling theorem. □

Finally, we are comfortable to drive the proof of the self-normalized concentration bounds.

*Proof of Theorem G.1.* Let $\Lambda = (\Lambda_t)$ be a sequence of independent Gaussian random variable with densities

$$f_{\Lambda_t}(\lambda) = c(L_t) \exp(-\frac{1}{2} L_t \lambda^2)$$

where $c(A) = \sqrt{A/2\pi}$ is a normalizing constant. We explicitly calculate $\bar{M}_t$ for any $t \geq 1$,

$$\bar{M}_t = \int_\mathbb{R} \exp\left(\lambda A_t - \frac{\lambda^2}{2} B_t^2\right) f_{\Lambda_t}(\lambda) d\lambda$$

$$= \int_\mathbb{R} \exp\left(-\frac{1}{2}\left(\lambda - \frac{A_t}{B_t^2}\right)^2 B_t^2 + \frac{1}{2}\frac{A_t^2}{B_t^2}\right) f_{\Lambda_t}(\lambda) d\lambda$$

$$= \exp\left(\frac{1}{2}\frac{A_t^2}{B_t^2}\right) \int_\mathbb{R} \exp\left(-\frac{1}{2}\left(\lambda - \frac{A_t}{B_t^2}\right)^2 B_t^2\right) f_{\Lambda_t}(\lambda) d\lambda$$

$$= c(L_t) \exp\left(\frac{1}{2}\frac{A_t^2}{B_t^2}\right) \int_\mathbb{R} \exp\left(-\frac{1}{2}\left(\left(\lambda - A_t/B_t^2\right)^2 B_t^2 + \lambda^2 L_t\right)\right) d\lambda.$$

Completing the square yields

$$\left(\lambda - \frac{A_t}{B_t^2}\right)^2 B_t^2 + \lambda^2 L_t = \left(\lambda - \frac{A_t}{L_t + B_t^2}\right)^2 (L_t + B_t^2) + \frac{A_t^2}{B_t^2} - \frac{A_t^2}{L_t + B_t^2}.$$

By the change of variables $\lambda' = \lambda - A_t/(L_t + B_t^2)$ in the following $(i)$,

$$\bar{M}_t = c(L_t) \exp\left(\frac{1}{2}\frac{A_t^2}{L_t + B_t^2}\right) \int_{\mathbb{R}} \exp\left(-\frac{1}{2}\left(\lambda - \frac{A_t}{L_t + B_t^2}\right)^2 (L_t + B_t^2)\right) d\lambda$$

$$\overset{(i)}{=} c(L_t) \exp\left(\frac{1}{2}\frac{A_t^2}{L_t + B_t^2}\right) \int_{\mathbb{R}} \exp\left(-\frac{1}{2}\left(\lambda^2(L_t + B_t^2)\right)\right) d\lambda$$

$$= \frac{c(L_t)}{c(L_t + B_t^2)} \exp\left(\frac{1}{2}\frac{A_t^2}{L_t + B_t^2}\right).$$

A final application of Markov's inequality yields

$$\mathbb{P}\left[|A_\tau| \geq \sqrt{2(L_\tau + B_\tau^2)\log\left(\frac{1}{\delta}\frac{(L_\tau + B_\tau^2)^{1/2}}{L_\tau^{1/2}}\right)}\right]$$

$$= \mathbb{P}\left[\frac{c(L_\tau)}{c(L_\tau + B_\tau^2)}\exp\left(\frac{1}{2}\frac{A_\tau^2}{L_\tau + B_\tau^2}\right) \geq \frac{1}{\delta}\right]$$

$$\leq \delta \cdot \mathbb{E}\left[\frac{c(L_\tau)}{c(L_\tau + B_\tau^2)}\exp\left(\frac{1}{2}\frac{A_\tau^2}{L_\tau + B_\tau^2}\right)\right]$$

$$\overset{i)}{\leq} \delta \cdot \mathbb{E}\left[\bar{M}_\tau\right] \overset{(ii)}{\leq} \delta,$$

where (i) uses the inequality for $\bar{M}_\tau$ derived above, and (ii) follows from Lemma G.8.

To get the anytime result in Theorem G.1, we define the stopping time

$$\tau = \min\left\{t \geq 1 : |A_t| \geq \sqrt{2(L_t + B_t^2)\log\left(\frac{1}{\delta}\frac{(L_t + B_t^2)^{1/2}}{L_t^{1/2}}\right)}\right\}$$

With an application of extended version of Lemma G.5, and applying the previous inequality yields

$$\mathbb{P}\left[\exists t \geq 1, |A_t| \geq \sqrt{2(L_t + B_t^2)\log\left(\frac{1}{\delta}\frac{(L_t + B_t^2)^{1/2}}{L_t^{1/2}}\right)}\right]$$

$$= \mathbb{P}\left[\tau < \infty, |A_\tau| \geq \sqrt{2(L_\tau + B_\tau^2)\log\left(\frac{1}{\delta}\frac{(L_\tau + B_\tau^2)^{1/2}}{L_\tau^{1/2}}\right)}\right]$$

$$\leq \mathbb{P}\left[|A_\tau| \geq \sqrt{2(L_\tau + B_\tau^2)\log\left(\frac{1}{\delta}\frac{(L_\tau + B_\tau^2)^{1/2}}{L_\tau^{1/2}}\right)}\right]$$

$$\leq \delta.$$

This completes the proof. □

## H  PROOF OF ARGUMENT FOR THE PRIOR MODEL

This section provide a new tool for random projection, dealing with the initial concentration in the prior model.

**Theorem H.1** (High-dimensional Hanson-Wright inequality). *Let $X_1, \ldots, X_n$ be independent, mean zero random vectors in $\mathbb{R}^M$, each $X_i$ is $K_i$-subGaussian. Let $A = (a_{ij})$ be an $n \times n$ matrix. Then for any $t \geq 0$, we have*

$$\mathbb{P}\left(|\sum_{i,j:i\neq j}^n a_{ij}\langle X_i, X_j\rangle| \geq t\right) \leq 2\exp\left(-\min\left\{\frac{t^2}{64K^4M\|A\|_F^2}, \frac{t}{8K^2\|A\|_2}\right\}\right)$$

*where $K = \max_i K_i$.*

*Remark* H.2. This is an high-dimension extension of famous Hanson-Wright inequality (Rudelson & Vershynin, 2013). The Theorem H.1 with exact constant is new in the literature, which maybe of independent interest. Our proof technique generalizes from (Rudelson & Vershynin, 2013).

**Lemma H.3** (Distributional JL lemma (Johnson & Lindenstrauss, 1984)). *For any $0 < \varepsilon, \delta < 1/2$ and $d \geq 1$ there exists a distribution $\mathcal{D}_{\varepsilon,\delta}$ on $\mathbb{R}^{M \times d}$ for $M = O\left(\varepsilon^{-2} \log(1/\delta)\right)$ such that for any $x \in \mathbb{R}^d$*

$$\mathbb{P}_{\Pi \sim \mathcal{D}_{\varepsilon,\delta}} \left( \|\Pi x\|_2^2 \notin \left[ (1-\varepsilon)\|x\|_2^2, (1+\varepsilon)\|x\|_2^2 \right] \right) < \delta$$

**Lemma H.4.** *We claim that the following construction of the random projection matrix $\Pi \in \mathbb{R}^{M \times d}$ with $M \geq 64\varepsilon^{-2} \log(1/\delta)$ satisfy the Lemma H.3: Let $\Pi = (\mathbf{z}_1, \ldots, \mathbf{z}_d)$ be a random matrix with each $\mathbf{z}_i \sim P_{\mathbf{z}}$, i.e., uniformly sampled over the unit sphere $\mathbb{S}^{M-1}$.*

*Proof.* Each $\mathbf{z}_i \sim P_{\mathbf{z}} = \mathcal{U}(\mathbb{S}^{M-1})$ is a $\frac{1}{\sqrt{M}}$-sub-Gaussian random vector with mean zero. Let $x \in \mathbb{R}^d$ be the vector to be projected. By the construction of $\Pi$,

$$\|\Pi x\|^2 - \|x\|^2 = \underbrace{\sum_{1 \leq i \neq j \leq d} x_i x_j \langle \mathbf{z}_i, \mathbf{z}_j \rangle}_{\text{off-diagonal}} + \underbrace{\sum_{i=1}^{d} x_i^2 (\|\mathbf{z}_i\|^2 - 1)}_{\text{diagonal}}$$

The diagonal term is zero due to the unit sphere $\mathbb{S}^{M-1}$. The JL lemma is then a consequence of bounding the following

$$\mathbb{P}_{\Pi} \left( |\text{off-diagonal}| \geq \varepsilon \|x\|^2 \right)$$

We apply Theorem H.1 with $A = xx^\top$ and $t = \varepsilon \|x\|^2$. Since $K = 1/\sqrt{M}$ and $\|A\|_F = \text{tr}(xx^\top) = \|x\|^2, \|A\|_2 = \|x\|^2$, then

$$\mathbb{P} \left( |\sum_{1 \leq i \neq j \leq d} x_i x_j \langle \mathbf{z}_i, \mathbf{z}_j \rangle| \geq \varepsilon \|x\|^2 \right) \leq 2 \exp \left( -\min \left\{ \frac{\varepsilon^2 \|x\|^4}{64 K^4 M \|A\|_F^2}, \frac{\varepsilon \|x\|^2}{8 K^2 \|A\|_2} \right\} \right)$$

$$\leq 2 \exp \left( -M \min \left\{ \varepsilon^2/64, \varepsilon/8 \right\} \right).$$

This implies that to get the RHS upper bound by $\delta$, we need

$$M \geq 64\varepsilon^{-2} \log(2/\delta).$$

$\square$

## H.1 PROOF OF THEOREM H.1

*Proof.* We prove the one-side inequality and the other side is similar by replacing $A$ with $-A$. Let

$$S = \sum_{i,j:i \neq j}^{n} a_{ij} \langle X_i, X_j \rangle. \tag{22}$$

**Step 1: decoupling.** Let $\iota_1, \ldots, \iota_d \in \{0, 1\}$ be symmetric Bernoulli random variables, (i.e., $\mathbb{P}(\iota_i = 0) = \mathbb{P}(\iota_i = 1) = 1/2$) that are independent of $X_1, \ldots, X_n$. Since

$$\mathbb{E}[\iota_i(1 - \iota_i)] = \begin{cases} 0, & i = j, \\ 1/4, & i \neq j, \end{cases}$$

we have $S = 4\mathbb{E}_\iota[S_\iota]$, where

$$S_\iota = \sum_{i,j=1}^{n} \iota_i(1 - \iota_j) a_{ij} \langle X_i, X_j \rangle$$

and the expectation $\mathbb{E}_\iota[\cdot]$ is the expectation taken with respect to the random variables $\iota_i$. By Jensen's inequality, we have

$$\mathbb{E}[\exp \lambda S] \leq \mathbb{E}_{X,\iota}[\exp 4\lambda S_\iota].$$

Let $\Lambda_\iota = \{i \in [d] : \iota_i = 1\}$. Then we write

$$S_\iota = \sum_{i \in \Lambda_\iota} \sum_{j \in \Lambda_\iota^c} a_{ij} \langle X_i, X_j \rangle = \sum_{j \in \Lambda_\iota^c} \langle \sum_{i \in \Lambda_\iota} a_{ij} X_i, X_j \rangle.$$

Taking expectation over $(X_j)_{j \in \Lambda_\iota^c}$ (i.e., conditioning on $(\iota_i)_{i=1,\dots,d}$ and $(X_i)_{i \in \Lambda_\iota}$), it follows that

$$\mathbb{E}_{(X_j)_{j \in \Lambda_\iota^c}}[\exp 4\lambda S_\iota] = \prod_{j \in \Lambda_\iota^c} \mathbb{E}_{(X_j)_{j \in \Lambda_\iota^c}}[\exp 4\lambda \langle \sum_{i \in \Lambda_\iota} a_{ij} X_i, X_j \rangle]$$

by the independence among $(X_j)_{j \in \Lambda_\iota}$. By the assumption that $X_i$ are independent sub-Gaussian with mean zero, we have

$$\mathbb{E}_{(X_j)_{j \in \Lambda_\iota^c}}[\exp 4\lambda S_\iota] \le \exp\left( \sum_{j \in \Lambda_\iota^c} 8\lambda^2 K_j^2 \| \sum_{i \in \Lambda_\iota} a_{ij} X_i \|^2 \right) =: \exp\left( 8\lambda^2 \sigma_\iota^2 \right).$$

Thus we get

$$\mathbb{E}_X[\exp 4\lambda S_\iota] \le \mathbb{E}_X[\exp 8\lambda^2 \sigma_\iota^2].$$

**Step 2: reduction to Gaussian random variables.** For $j = 1, \dots, n$, let $g_j$ be independent $N\left(0, 16K_j^2 I\right)$ random variables in $\mathbb{R}^M$ that are independent of $X_1, \dots, X_n$ and $\iota_1, \dots, \iota_n$. Define

$$T := \sum_{j \in \Lambda_\iota^c} \langle g_j, \sum_{i \in \Lambda_\iota} a_{ij} X_i \rangle.$$

Then, by the definition of Gaussian random variables in $\mathbb{R}^M$, we have

$$\mathbb{E}_g[e^{\lambda T}] = \prod_{j \in \Lambda_\iota^c} \mathbb{E}_g[\exp \langle g_j, \lambda \sum_{i \in \Lambda_\iota} a_{ij} X_i \rangle]$$

$$= \exp\left( 8\lambda^2 \sum_{j \in \Lambda_\iota^c} K_j^2 \| \sum_{i \in \Lambda_\iota} a_{ij} X_i \|^2 \right) = \exp\left( 8\lambda^2 \sigma_\iota^2 \right)$$

So it follows that

$$\mathbb{E}_X[\exp 4\lambda S_\iota] \le \mathbb{E}_{X,g}[\exp \lambda T].$$

Since $T = \sum_{i \in \Lambda_\iota} \langle \sum_{j \in \Lambda_\iota^c} a_{ij} g_j, X_i \rangle$, by the assumption that $X_i$ are independent sub-Gaussian with mean zero, we have

$$\mathbb{E}_{(X_i)_{i \in \Lambda_\iota}}[\exp \lambda T] \le \exp\left( \frac{\lambda^2}{2} \sum_{i \in \Lambda_\iota} K_i^2 \| \sum_{j \in \Lambda_\iota^c} a_{ij} g_j \|^2 \right),$$

which implies that

$$\mathbb{E}_X[\exp 4\lambda S_\iota] \le \mathbb{E}_g[\exp\left( \lambda^2 \tau_\iota^2 / 2 \right)] \tag{23}$$

where $\tau_\iota^2 = \sum_{i \in \Lambda_\iota} K_i^2 \| \sum_{j \in \Lambda_\iota^c} a_{ij} g_j \|^2$. Note that $\tau_\iota^2$ is a random variable that depends on $(\iota_i)_{i=1}^d$ and $(g_j)_{j=1}^n$.

**Step 3: diagonalization.** We have $g_j = \sum_{k=1}^M \langle g_j, e_k \rangle e_k$ and

$$\tau_\iota^2 = \sum_{i \in \Lambda_\iota} K_i^2 \left\| \sum_{j \in \Lambda_\iota^c} a_{ij} g_j \right\|^2 = \sum_{i \in \Lambda_\iota} K_i^2 \left\| \sum_{k=1}^M \left( \sum_{j \in \Lambda_\iota^c} a_{ij} \langle g_j, e_k \rangle \right) e_k \right\|^2$$

$$= \sum_{k=1}^M \sum_{i \in \Lambda_\iota} \left( \sum_{j \in \Lambda_\iota^c} K_i a_{ij} \langle g_j, e_k \rangle \right)^2$$

$$= \sum_{k=1}^M \| P_\iota \tilde{A} (I - P_\iota) G_k \|^2$$

where the last second step follows from Parseval's identity. $G_{jk} := \langle g_j, e_k \rangle$, $j = 1, \ldots, n$, are independent $N\left(0, 16K_j^2\right)$ random variables. $G_k = (G_{1k}, \ldots, G_{nk})^\top \in \mathbb{R}^n$. $\tilde{A} = (\tilde{a}_{ij})_{i,j=1}^n$ with $\tilde{a}_{ij} = K_i a_{ij}$. Let $P_\iota \in \mathbb{R}^{n \times n}$ be the restriction matrix such that $P_{\iota,ii} = 1$ if $i \in \Lambda_\iota$ and $P_{\iota,ij} = 0$ otherwise.

Define normal random variables $Z_k = (Z_{1k}, \ldots, Z_{nk})^\top \sim N(0, I)$ for each $k = 1, \ldots, M$. Then we have $G_k \overset{D}{=} \Gamma^{1/2} Z_k$ where $\Gamma = 16 \operatorname{diag}(K_1^2, \ldots, K_n^2)$.

Let $\tilde{A}_\iota := P_\iota \tilde{A}(I - P_\iota)$. Then by the rotational invariance of Gaussian distributions, we have

$$\sum_{k=1}^M \|\tilde{A}_\iota G_k\|^2 \overset{D}{=} \sum_{k=1}^M \|\tilde{A}_\iota \Gamma^{1/2} Z_k\|^2 \overset{D}{=} \sum_{k=1}^M \sum_{j=1}^n s_j^2 Z_{jk}^2$$

where $s_j^2, j = 1, 2, \ldots, n$ are the eigenvalues of $\Gamma^{1/2} \tilde{A}_\iota^\top \tilde{A}_\iota \Gamma^{1/2}$.

**Step 4: bound the eigenvalues.** It follows that

$$\max_{j \in [n]} s_j^2 = \|\tilde{A}_\iota \Gamma^{1/2}\|_{\operatorname{op}}^2 \leq 16 K^4 \|A\|_2^2.$$

In addition, we also have

$$\sum_{j=1}^n s_j^2 = \operatorname{tr}(\Gamma^{1/2} \tilde{A}_\iota^\top \tilde{A}_\iota \Gamma^{1/2}) \leq 16 K^4 \|A\|_F^2$$

and $\sum_{k=1}^M \sum_{j=1}^n s_j^2 \leq 16 M K^4 \|A\|_F^2$. Invoking Equation (23), we get

$$\mathbb{E}_X\left[e^{4\lambda S_\iota}\right] \leq \prod_{k=1}^M \prod_{j=1}^n \mathbb{E}_Z\left[\exp\left(\lambda^2 s_j^2 Z_{jk}^2/2\right)\right]$$

Since $Z_{jk}^2$ are i.i.d. $\chi_1^2$ random variables with the moment generating function $\mathbb{E}[e^{tZ_{jk}^2}] = (1 - 2t)^{-1/2}$ for $t < 1/2$, we have

$$\mathbb{E}_X\left[e^{4\lambda S_\iota}\right] \leq \prod_{k=1}^M \prod_{j=1}^n \frac{1}{\sqrt{1 - \lambda^2 s_j^2}} \quad \text{if } \max_j \lambda^2 s_j^2 < 1.$$

Using $(1 - z)^{-1/2} \leq e^z$ for $z \in [0, 1/2]$, we get that if $16 K^4 \|A\|_2^2 \lambda^2 < 1$, then

$$\mathbb{E}_X\left[e^{4\lambda S_\iota}\right] \leq \exp\left(\lambda^2 \sum_{k=1}^M \sum_{j=1}^n s_j^2\right) \leq \exp\left(16 \lambda^2 K^4 \|A\|_F^2\right).$$

Note that the last inequality is uniform in $\iota$. Taking expectation with respect to $\delta$, we obtain that

$$\mathbb{E}_X\left[e^{\lambda S}\right] \leq \mathbb{E}_{X,\iota}\left[e^{4\lambda S_\iota}\right] \leq \exp\left(16 \lambda^2 M K^4 \|A\|_F^2\right)$$

whenever $0 < \lambda < (4K^2\|A\|_2)^{-1}$.

**Step 5: Conclusion.** Step 5: conclusion. Now we have

$$\mathbb{P}(S \geq t) \leq \exp\left(-\lambda t + 16\lambda^2 M K^4 \|A\|_F^2\right) \quad \text{for } 0 < \lambda \leq \left(4K^2\|A\|_2\right)^{-1}$$

Optimizing in $\lambda$, we deduce that there exists a universal constant $C > 0$ such that

$$\mathbb{P}(S \geq t) \leq \exp\left[-\min\left(\frac{t^2}{64 M K^4 \|A\|_F^2}, \frac{t}{8K^2\|A\|_2}\right)\right].$$

$\square$

## I  VERIFY THE ASSUMPTION FOR SOME TYPICAL DISTRIBUTIONS

**Lemma I.1** (MGF of Beta distribution). *For any $\alpha, \beta \in \mathbb{R}_+$ with $\alpha \leq \beta$. Random variable $X \sim \text{Beta}(\alpha, \beta)$ has variance $\text{Var}(X) = \frac{\alpha\beta}{(\alpha+\beta)^2(\alpha+\beta+1)}$ and the centered MGF $\mathbb{E}[\exp(\lambda(X - \mathbb{E}[X]))] \leq \exp(\frac{\lambda^2 \text{Var}(X)}{2})$.*

**Example I.2** (Uniform over sphere $\mathcal{U}(\mathbb{S}^{M-1})$). Given a random vector $\mathbf{z} \sim \mathcal{U}(\mathbb{S}^{M-1})$, for any $v \in \mathbb{S}^{M-1}$, we have

$$\langle \mathbf{z}, v \rangle \sim 2\,\text{Beta}\left(\frac{M-1}{2}, \frac{M-1}{2}\right) - 1.$$

Thus, by Lemma I.1, we confirm $\mathbf{z}$ is $(1/\sqrt{M})$-sub-Gaussian random vector.

*Proof of Lemma I.1.* For $X \sim \text{Beta}(\alpha, \beta)$, Skorski (2023) gives a novel Order 2 Recurrence for Central Moments.

$$\mathbb{E}\left[(X - \mathbb{E}[X])^p\right] = \frac{(p-1)(\beta - \alpha)}{(\alpha+\beta)(\alpha+\beta+p-1)} \cdot \mathbb{E}\left[(X - \mathbb{E}[X])^{p-1}\right]$$
$$+ \frac{(p-1)\alpha\beta}{(\alpha+\beta)^2(\alpha+\beta+p-1)} \cdot \mathbb{E}\left[(X - \mathbb{E}[X])^{p-2}\right]$$

Let $m_p := \frac{\mathbb{E}[(X-\mathbb{E}[X])^p]}{p!}$, When $\alpha \leq \beta$, it follows that $m_p$ is non-negative when $p$ is even, and negative otherwise. Thus, for even $p$,

$$m_p \leq \frac{1}{p} \cdot \frac{\alpha\beta}{(\alpha+\beta)^2(\alpha+\beta+p-1)} m_{p-2} \leq \frac{\text{Var}(X)}{p} \cdot m_{p-2}.$$

Repeating this $p/2$ times and combining with $m_p \leqslant 0$ for odd $p$, we obtain

$$m_p \leqslant \begin{cases} \frac{\text{Var}(X)^{\frac{p}{2}}}{p!!} & p \text{ even} \\ 0 & d \text{ odd} \end{cases}.$$

Using $p!! = 2^{p/2}(p/2)!$ for even $p$, for $t \geqslant 0$ we obtain

$$\mathbb{E}[\exp(\lambda[X - \mathbb{E}[X]])] \leqslant 1 + \sum_{p=2}^{+\infty} m_p \lambda^p = 1 + \sum_{p=1}^{+\infty} (\lambda^2 \text{Var}(X)/2)^p / p! = \exp\left(\frac{\lambda^2 \text{Var}(X)}{2}\right)$$

$\square$

