# OpenReview forum: "Efficient and scalable reinforcement learning via hypermodel"
_ICLR.cc/2024/Conference — ICLR 2024 Conference Withdrawn Submission_

### Official Review · Reviewer_87xX · 2023-10-15

**Soundness:** 4 excellent
**Presentation:** 3 good
**Contribution:** 4 excellent
**Rating:** 6
**Confidence:** 4

**Summary:**

Claim is to provide an efficient hypermodel for posterior sampling for Thompson sampling to guide deep exploration. Claim is to be the first to provide results in deep Atari problems.

**Strengths:**

Writing style is precise.

Arguments are convincing.

Results are impressive.

**Weaknesses:**

English lacks particles.

Section 5 should be rewritten. Should provide more intuitive explanation why the algorithm is better. Take more explanation of Appendix D to main paper.
No Conclusion or Discussion Section.

**Questions:**

Can you add a conclusion? Can you please enhance the intuitive explanation of how the algorithm works?

The authors should provide code, in a GitHub link, otherwise not reproducible

---

> ### Comment · Reviewer_87xX · 2023-11-22
>
> After reading the other reviews I have adjusted my score. The reviews pointed out weaknesses that are important.

---

> ### Author Response · Authors · 2023-11-29
> **Response to Reviewer 87xX**
>
> Dear reviewer,
>
> Thanks for your time and comments!
>
> ### Answer to Q1
> Please take a look on the revised section 5
>
> ### Open source
> Our code, scripts for all experiments and results on Atari games have been released in [https://anonymous.4open.science/r/HyperFQI-D277](https://anonymous.4open.science/r/HyperFQI-D277)

---

### Official Review · Reviewer_bSoB · 2023-10-19

**Soundness:** 1 poor
**Presentation:** 2 fair
**Contribution:** 2 fair
**Rating:** 3
**Confidence:** 5

**Summary:**

The paper proposes HyperFQI, an approximate Thompson Sampling method for RL using hypermodel as backbone of it for uncertainty estimation of action-value function. Experiments are provided for the DeepSea environment as well as Atari suite.

**Strengths:**

The algorithm HyperFQI is practically scalable and computationally efficient. The analysis in the DeepSea environment is thorough and detailed. The Atari experiments are also promising.

**Weaknesses:**

Unfortunately, the work ignores many of the relevant prior works and failed to cite them/compare HyperFQI against them. In the abstract, the paper says that “this is the first principled RL algorithm that is provably efficient and also practically scalable to complex environments such as Arcade learning environment.” But this is not true.  LSVI-PHE (Ishfaq et al. 2021), LMC-LSVI (Ishfaq et al 2023),  BayesUCBVI algorithm from Tiapkin et. al, ICML 2022 are examples of TS based methods that are both provably efficient and practically scalable. The work also doesn’t compare HyperFQI against these algorithms in the experiment section. Consequently, many of the “novelty” and “first one achieving” type of claims seem to be over-claim.

The Atari experiment uses only 26 games and uses 2M training data (i.e. 10M frames) whereas the standard is to use 200M frames. At the very least, I would expect the authors to include results of 50M frames. The results provided in Table 4 is very skewed and unfair representation of performance for the baselines. Also it’s not clear how the authors found numbers for DDQN and Rainbow from Hessel et al. (2018) for 2M training data given the tables in that paper represent results for 200M training frames (Please see Question 14). Given 200M training frames, most of the baselines seem to perform SIGNIFICANTLY better than reported result based on Hessel et al. (2018). So, I think it is imperative that HyperFQI be trained for larger number of training frames for better and fair comparison.

It is also not clear what was the basis of picking these particular 26 games from Atari 57 suite. Even though the paper mentions Bellemare et al. (2016) as a basis for picking easy exploration, hard exploration (dense reward) and hard exploration (sparse reward), the paper uses only a single hard exploration (sparse reward) task namely Freeway in the experiment whereas Bellemare et al. (2016) classified 7 games as sparse reward hard exploration task. Could you provide results for those games for better and fair comparison? In particular, could you provide result for Pitfall, Gravitar, Solaris and Venture as some of the baselines have been shown to work well for these games?

Regarding the regret bound result, first and foremost a detailed proof is missing and the paper only mentions some lemmas and the final theorem without any proof (both in the main paper and the appendix).

Typos:
In section 2.1, “terminal $\in \mathcal{S}$”
In Section 2.2, the definition of $f_\theta$ for ensemble model seems to have typo as well.
The last line of Section 2.2 “importance of difference” —> “important differences”

**Questions:**

1. In Figure 1, why was the comparison made using only 26 Atari games? What was the basis of choosing those games? I would expect someone to use all 57 games or just use the hardest exploration tasks for fair comparison.

2. In the introduction, what did you mean by “randomly perturbing  a prior”?

3. In equation 2, what is the difference between the fixed prior model and randomized prior function proposed in Osband et al 2018?

4. > HyperFQI selects the action based on sampling single or multiple indices and then taking the action with the highest value from hypermodels applying these indices. This can be viewed as an value-based approximate (optimistic) Thompson sampling via Hypermodel.

What is the difference between this version of HyperFQI and LSVI-PHE proposed in Ishfaq et al. 2021?

5. What is the difference between $P_z$ and $P_\xi$?

6. What is the role of $\sigma$ ins Eq (3)?

7. What is $l^{\gamma, \sigma}_{z}$? It’s not defined.

8. In Eq (5), what is the role of $|D|/|\tilde{D}|$?

9. For the experiments, do you choose the hyperparameters based of the different seeds from the ones you use for evaluation?

10. How do you define solving the DeepSea environment in the presented experiments?

11. Did you try BootDQN with randomized prior function (Osband et al 2018) as a baseline in Fig 2?
12. Given the optimistic sampling strategy of HyperFQI-OIS is very similar to LSVI-PHE proposed in Ishfaq et al 2021, it is important to include it as a baseline for the DeepSea experiment. Can you compare your algorithm against it?

13. What is NpS mentioned under “ablation study” of Section 4.1?

14. > In Table 4, we present the best score achieved in each environment with 2M steps. The scores for Rainbow and DDQN are obtained from Hessel et al. (2018)

I checked Hessel et al. (2018) and couldn’t find a table that has result for 2M steps. The tables are for 200M frames results. How did you find the 2M step results for Rainbow and DDQN from Hessel et al. (2018)?

15. Is it possible to have an implementation of the algorithms in supplemental material? Knowing that the contribution of the paper is both theoretical and computational, it seems important to me to have a public anonymized code.


Ishfaq, H., Cui, Q., Nguyen, V., Ayoub, A., Yang, Z., Wang, Z., Precup, D. and Yang, L., 2021, July. Randomized exploration in reinforcement learning with general value function approximation. In International Conference on Machine Learning

Ishfaq, H., Lan, Q., Xu, P., Mahmood, A.R., Precup, D., Anandkumar, A. and Azizzadenesheli, K., 2023. Provable and Practical: Efficient Exploration in Reinforcement Learning via Langevin Monte Carlo. arXiv preprint arXiv:2305.18246.

Tiapkin D, Belomestny D, Moulines É, Naumov A, Samsonov S, Tang Y, Valko M, Ménard P. From Dirichlet to Rubin: Optimistic exploration in RL without bonuses.

---

> ### Author Response · Authors · 2023-11-29
> **Response to Reviewer bSoB**
>
> Dear reviewer,
>
> Thanks for your time and comments!
>
> ### Answer for Q1
> We did not choose the 26 Atari games ourselves, but instead followed the established practice in popular and widely accepted research (Van Hasselt et al., 2019)  (Ye et al., 2021).
> By using these 26 Atari games to evaluate our HyperFQI.
> Additionally, we tested our HyperFQI on the 8 most hardest exploration tasks, and the results are presented in Figure 15 from Appendix C.2 .
> Due to time and resource constraints, we could only train our HyperFQI for up to 2 million steps.
> Nevertheless, we observed that our HyperFQI outperformed other baselines in the majority of games, providing strong evidence for its superiority.
>
> ### Answer for Q2
> "randomly perturbing the prior" means the RLSVI algorithm samples the independent $\hat{\theta}_{0}$ from prior distribution.
>
> ### Answer for Q3
> The fixed prior model will produce an i.i.d sample from prior distribution with high probability. We use a lower dimensional source of randomness to generate the model, i.e. M-> d where M is small. randomized prior function proposed in Osband et al 2018 can be thought as a special case of our fixed prior model when the index distribution is uniform distribution over one-hot vectors.
>
> ### Answer to Q4 and Q5
> In hyperFQI, we use $P_{\xi}$ as multivariate gaussian distribution. and $P_{z}$ is uniform distribution over sphere.
> Ensemble based methods are special case of hypermodel when $P_{\xi}$ is uniform distribution over $\{e_1, \ldots, e_M\}$. We refer it to HyperFQI-OH meaning one-hot index.
> Therefore, LSVI-PHE, in principle, can be regarded as a special case of HyperFQI-OH with OIS.
> Compared to one-hot index, Gaussian index allows us to sample infinitely different index, allowing better (optimistic) exploration and leading to final performance. This is justified in the DeepSea environments.
>
> As we were unable to find the official implementation of LSVI-PHE, we applied the OIS to BootDQN with prior to represent the LSVI-PHE  as the implementation guided by the original article cite (Ishfaq et al. 2021)
>
> ### Ans to Q6
> This is for controlling the noise level of injected noise. detailed explanation in revision.
>
> ### Ans to Q7
> typo. revised.
>
> ### Answer to Q8
> See revised version. It is for monte carlo approximation of the expectation in equation 4.
>
> ### Answer to Q9
> No, we do not pick seeds based on the result of evalutation results. We use the same 20 seeds for evaluation for each environments in Atari benchmarks.
>
> ### Answer to Q10
> Please take a look at the revision.
>
> ### Answer to Q11
> We actually tried BootDQN with randomized prior function (Osband et al 2018). There is a typo in the citation of original submission.
>
> ### Answer to Q11 & Q12
> We utilized the BootDQN with prior  by following the guidelines provided in [https://github.com/google-deepmind/bsuite](https://github.com/google-deepmind/bsuite). As we were unable to find the official implementation of LSVI-PHE, we applied the OIS to BootDQN with prior to represent the LSVI-PHE  as the implementation guided by the original article cite (Ishfaq et al. 2021). For both bootstrapped methods, we firstly set the number of ensembles to 4, consistent with the dimension of index used in HyperFQI. As depicted in the Figure 9(a) from Appendix C.1, BootDQN w. prior fails to solve DeepSea with size 20 within $10e3$ episodes. Although OIS enhances exploration, BootDQN-OIS w. prior knowledge solves DeepSea with size 20 with in few seeds but fails in DeepSea with size 30. Furthermore, we compared our HyperFQI method to bootstrapped methods using 16 ensemble networks, as illustrated in Figure 9(b) from Appendix C.1. While increasing the number of ensembles improves exploration, it remains insufficient for solving DeepSea with large size. In contrast, our HyperFQI approach efficiently solves DeepSea with just 4 dimensions of index, demonstrating its superiority.
>
> ### Answer to Q13 notation consistency
> Please take a look on the revised version. Now we revise it as $|\tilde{\Xi}|$ for consistency, first appeared in equation 5, representating the number indices to approximate the expectation.
>
> ### Answer to Q14
> We utilized the training results of Rainbow and DQN from  [https://github.com/google-deepmind/dqn_zoo/blob/master/results.tar.gz](https://github.com/google-deepmind/dqn_zoo/blob/master/results.tar.gz)  , which were based on 200M Frames. Specifically, we extracted the first 20M steps from these results to compare them with our HyperFQI. Additionally, we obtained the original results for HyperDQN from the authors and similarly extracted the first 20M steps for comparison purposes.
>
> ### Answer to Q15
> Our code, scripts for all experiments and results on Atari games have been released in [https://anonymous.4open.science/r/HyperFQI-D277](https://anonymous.4open.science/r/HyperFQI-D277)

---

### Official Review · Reviewer_h2ut · 2023-10-31

**Soundness:** 3 good
**Presentation:** 4 excellent
**Contribution:** 3 good
**Rating:** 6
**Confidence:** 3

**Summary:**

This paper proposes a scalable reinforcement learning algorithm capable of performing deep exploration based on hypermodels. The proposed HyperFQI algorithm samples a random index from a reference distribution, then computes an approximate posterior based on the drawn index. This index is meant to capture the epistemic uncertainty of the agent. HyperFQI is shown to perform deep exploration on the deepsea benchmark and achieving super-human performance on Atari benchmarks. A sublinear regret bound is proved for HyperFQI under a tabular MDP.

**Strengths:**

This paper provides convincing experiment results for their proposed algorithm. Even though the algorithm carries over the spirit of hypermodels [1], the regret bound seems novel to me. Overall, a very well-written paper with clear demonstrations and important messages.

[1] Hypermodels for Exploration. Vikranth Dwaracherla, Xiuyuan Lu, Morteza Ibrahimi, Ian Osband, Zheng Wen, Benjamin Van Roy. ICLR 2020.

**Weaknesses:**

Some experiment results are not explained very well. See questions below.

**Questions:**

I'm not sure how to interpret Figure 3. The trends in the left and right graphs seem to be opposite in $M$. Could you explain why this is the case?

The analysis asssumes an independent Dirichlet prior over transitions, which is a rather strong assumption. Can this assumption be relaxed? Does the analysis heavily rely on this assumption?

What is "Assumption 3" in the statement of Theorem 5.5? I am assuming it's Assumption 5.1. Please use hyperref rather than hardcoding the theorem numbers.

---

> ### Author Response · Authors · 2023-11-29
> **Response to Reviewer h2ut**
>
> Dear reviewer,
>
> Thanks for your time and comments!
>
> ### Explanation of Experimental Results (Question on Figure 3):
> Please take a look at the revised version for detailed explanation.
>
> ### Question on analysis assumption
> This is discussed in Remark 5.4 in the follow of the theorem.
> Also the assumption can be relaxed and it seems straightforward to generalize our Bayesian analysis to worse-case regret analysis as long as using our established key lemma 5.2.
>
> ### Discussion on the literature [1]
> From theoretical side, [1] do not provide any justification on the posterior approximation capability of hypermodel nor regret analysis even for bandit problem.
> From practical side, we describe a lot of difference with [1] and the difference with some other related work in introduction and appendix A.1.3.
>
> [1] Hypermodels for Exploration. Vikranth Dwaracherla, Xiuyuan Lu, Morteza Ibrahimi, Ian Osband, Zheng Wen, Benjamin Van Roy. ICLR 2020

---

### Official Review · Reviewer_47P5 · 2023-11-01

**Soundness:** 1 poor
**Presentation:** 1 poor
**Contribution:** 2 fair
**Rating:** 3
**Confidence:** 5

**Summary:**

The paper proposes HyperFQI, which uses hypermodels to approximate the epistemic uncertainty of the Q-value model in a tractable manner. The resulting hyper model allows approximate Thompson sampling for reinforcement learning. On Atari benchmark, the proposed method is able to achieve human-level performance with a small parameter count. The authors also provide a theoretical regret bound for HyperFQI under the tabular case with the finite-horizon time-inhomogeneous MDP assumption.

**Strengths:**

The algorithm is simple and takes an important step towards developing a practical algorithm that approximates the epistemic uncertainty of the Q-function, which is very important for efficient online exploration and learning.

**Weaknesses:**

While the method seems novel and principled, there are two big weaknesses of the paper that stand out to me.

The first is the lack of empirical evaluations is one of the biggest weaknesses of the paper. See below:
- The authors put a big emphasis on the number of parameters that the method needs for the Atari benchmark (e.g., Figure 1), and show that the method is parameter-efficient. While parameter count is an interesting metric to study, it is not convincing to me that the method would be able to scale up.
- Are there experiments that show that having large index dim helps on Atari (e.g.., similar to the ablation study shown in Figure 5)? Without that it is unclear to me whether the results on Atari is due to better parameter tuning or due to the proposed method.

The second weakness is the theory section (Section 5). The authors list a bunch of results in the main body, but I could not find proofs to any of them anywhere in the paper. Also, there are lot of missing pieces that could have made the theoretical results much better positioned in the literature. A few examples,
- Section 5.1: Why is finite-horizon time-inhomogeneous MDP a reasonable assumption to make? What are some related works that also analyzed under this setting and how do their results compare?
- Assumption 5.1: Is the Dirichlet prior assumption important for the results? Do prior works also make the same assumption?

**Questions:**

- What incentivizes the model to pay attention to the noise? For example, the hypermodel could degenerate to be a normal Q-network
- Equation (5) — is $\xi^-$ being sampled from $P_\xi$ every time the loss is being optimized?

---

> ### Author Response · Authors · 2023-11-29
> **Response to Reviewer 47P5**
>
> Dear reviewer,
>
> Thanks for your time and comments!
>
> ### Response to W1
> As described in Section 4, the scalability of our HyperFQI does not imply that our network can be extended to large-scale. Instead, our scalability is intended to address tasks with complex state spaces using a network of minimal size, i.e., lower computation costs. Our HyperFQI, with only 4 index dimensions, can successfully solve DeepSea problems ranging from size 20 to 100. It can even achieve human-level performance in Atari games with pixel observations, requiring fewer intersections.  All empirical results demonstrate the efficiency and scalability of our HyperFQI .
>
> Figure 5 is not about increasing index dimension or increasing model size. Please take a look for the revised version for detailed explanation.
>
>
> Due to the high cost involved, we do not perform parameter tuning on Atari games. Nevertheless, we ran DQN with the same network architecture and hyper-parameters for comparison, referred to as DDQN(ours) in Table 2. Our results show that HyperFQI significantly outperforms DDQN(ours), which demonstrates the advantage of our method.
>
> ### Response to W2
> The assumption issue is discussed in Remark 5.4 in the follow of the theorem.
> Full details of theoretical results is included.
>
> ### Response to Q1
> Please refer to the section 5 for illustrative idea.
> For one-sentence intuitive understanding, the state-action pairs that lacks data in the buffer will incur high variance for the value function on such state-action. The high variance will be backup to initial state step by step, incentivising a deep exploration behvior to the state rarely visited.
>
> ### Response to Q2
> We have revised the equation 5 and explain the objective carefully.
> We also explain its practical implementation in the appendix A.1.2.
> For continuous or uncountable infinite state space, we just resample independent $\xi^-$ for each data tuple d in the mini-batch.

---

### Official Review · Reviewer_2VR8 · 2023-11-02

**Soundness:** 2 fair
**Presentation:** 3 good
**Contribution:** 3 good
**Rating:** 5
**Confidence:** 3

**Summary:**

This paper introduces HyperFQI, a framework for inducing enhanced exploration using Thompson sampling in reinforcement learning. The paper approximates the posterior distribution of action-values with a hypermodel formulation and selects an action by sampling a noisy index from a fixed distribution, then combines it with the learned value function. These perturbations, caused by the noisy sample, motivate the agent to explore more and identify interesting states. When tested on the DeepSea benchmark and 26 Atari games (2M interactions), results indicate HyperFQI surpasses baselines like Rainbow and HyperDQN.

**Strengths:**

S1: The paper is well-written and comprehensible for the most part. Data presented in tables and graphs effectively highlight the improvements over the baselines.

S2: Employing a hypermodel to approximate the posterior distribution leads to an efficient algorithm with minimal computational overhead.

S3: The conducted experiments are thorough and provide statistically significant evidence of HyperFQI's strengths (though refer to W2 regarding the Atari games).

**Weaknesses:**

W1: Although the paper is well-constructed, its primary motivation — efficiently approximating the posterior distribution for Thompson sampling—is not novel. Specifically, [1] presents a similar idea, opting for a variational distribution to approximate the posterior in complex observation environments. The primary distinction seems to be the choice between hypermodel and variational distribution.

W2: The mentioned best scores for the Atari games appear to represent best scores from the 20 seeds. I believe that standard reporting typically involves the average score across these seeds rather than cherrypicking only the top scores.

W3: The experimental methodology, aiming to demonstrate advantages, is not entirely balanced. The paper employs different architectures (DQN nature vs Rainbow vs HyperFQI) and learning algorithms (Fitted Q-iteration vs Q-learning) when compared with the baselines. Additionally, training the model on 2M interactions doesn't exactly highlight data efficiency. A 100K interaction scenario might have been more appropriate.

[1]  Aravindan, Siddharth, and Wee Sun Lee. 2021. “State-Aware Variational Thompson Sampling for Deep Q-Networks.” arXiv [Cs.LG]. arXiv. http://arxiv.org/abs/2102.03719.

**Questions:**

Q1: Can the authors provide clarity on their methodology for comparing scores obtained on Atari games?

Q2: Would the authors care to address the comparison with the paper referenced in W1?

---

> ### Author Response · Authors · 2023-11-29
> **Response to reviewer 2VR8**
>
> Dear reviewer,
>
> Thanks for your time and comments!
>
> ### Response to W1 and Q2
>
> First, thank you for pointing out the literature. Computation-efficient posterior approximation or approximate bayesian inference is a long-standing question for whole community, both deep learning and reinforcement learning.
>
> We provide a way with theoretical justification to approximate the posterior. The core idea for approximating posterior in hypermodel is describe in lemma 5.2. We provide a new probability tool to justify its approximation correctness and computation efficiency. This is new to the approximate posterior sampling reinforcement learning literature.
>
> Secondly, we have adopted SANE as our new baseline and utilized their official implementation from https://github.com/NUS-LID/SANE to reproduce the results. We conduct the experiments on 8 hardest exploration Atari games, using 2 million steps, and the results are presented in Figure 15 from Appendix C.2. Our HyperFQI outperforms SANE significantly.
>
> ### Response to W2 and Q1
>
> We provide a more comprehensive explanation of our experimental protocol in Appendix C.2. We don not select the best score across seeds.
>
> Our experimental protocol is the same as the our baseline HyperDQN (Li et al., 2022a) as well as some other well-known literatures, e.g. (Mnih et al., 2015) and and (van Hasselt et al., 2016).
>
> The protocol includes 1) for each Atari game, we run 20 seeds in total 2) each seed will produce one best model in hindsight, leading to 20 different models 3) we evaluate 20 models, each for 200 times.
>
> 4) We then calculate the average score from these 200 evaluations as the score for each model associated with each seed.
>
> 5) Finally, we calculate and report the average score across 20 seeds as the final score for each Atari game.
>
> ### Response to W3
>
> For reproducing results in Atari games, we suggest using the developers' own results or implementation codes since they have likely released their best possible outcomes. Our baselines utilize state-of-the-art value-based algorithms, showcasing the superiority of our HyperFQI. Moreover, we ran Double DQN with the same network architecture and hyper-parameters for comparison, referred to as DDQN(ours) in Table 1. Our results show that HyperFQI significantly outperforms DDQN(ours), which demonstrates the advantage of our method.
>
> Furthermore, our experimental results on DeepSea further demonstrate the advantages of our HyperFQI, as all of our baselines used the same hyper-parameters and feature extraction network. Our HyperFQI achieved the best performance and significantly outperformed our other baselines.
>
> In addition. our HyperFQI can significantly outperform the DER (Van Hasselt et al., 2019) on the Atari100K benchmark, as demonstrated through comparison results. To assess DER's performance using the Atari100K benchmark, we utilized the widely-used implementation available on https://github.com/Kaixhin/Rainbow, which can produce results comparable to those of the original paper. These findings also demonstrate the data efficiency of our HyperFQI approach.
>
> | Method | IQM | Mean | Median |
>
> | :--------| :----------------:| :---------------: | :--------------: |
>
> |DER | 0.17 (0.15, 0.18) | 0.33 (0.30, 0.37) | 0.19 (0.17, 0.21)|
>
> |HyperFQI | 0.20 (0.19, 0.21) | 0.43 (0.41, 0.45) | 0.19 (0.16, 0.21)|

---

### Author Response · Authors · 2023-11-22
**Open Source HyperFQI**

Dear reviewers,

You can find the repository for our proposed algorithm. We give the scripts with which anyone can easily reproduce our results.
https://anonymous.4open.science/r/HyperFQI-D277
We believe that this address the concerns of most reviewers.

Best regards!

---

### Author Response · Authors · 2023-11-22
**General Response**

Dear reviewers,

Thank you for help leading this work to the next level!
We have made revisions concerning your questions and comments. The revised part are coloured with red colour.

We have done all comparison experiments that the reviewers asked, including the additional comparison with LSVI-PHE (Ishfaq et al. 2021), LMC-LSVI (Ishfaq et al 2023) and SANE (Aravindan et al 2021). The additional results can be found in appendix C with annotations and explanations.

We have included the full details of theoretical results. There are some technical contributions which maybe of independent interest. The most important technical contribution is a novel probability tool for sequential random projection that enables our regret analysis. This probability tool is new in the literature of random projection and also sequential analysis, which can be found in appendix G and maybe of independent interest.

We have also refined the explanation of our proposed HyperFQI with simple illustrations.

Please have a look on the revised paper we uploaded!

Thank you for your time!